# Improving the Variance of Differentially Private Randomized Experiments through Clustering

**Adel Javanmard** [1,2] **Vahab Mirrokni** [2] **Jean Pouget-Abadie** [2]

## Abstract

Estimating causal effects from randomized experiments is only possible if participants are willing to disclose their potentially sensitive responses. Differential privacy, a widely used framework for ensuring an algorithm's privacy guarantees, can encourage participants to share their responses without the risk of de-anonymization. However, many mechanisms achieve differential privacy by adding noise to the original dataset, which reduces the precision of causal effect estimation. This introduces a fundamental trade-off between privacy and variance when performing causal analyses on differentially private data. In this work, we propose a new differentially private mechanism, CLUSTER-DP, which leverages a given cluster structure in the data to improve the privacy-variance trade-off. While our results apply to any clustering, we demonstrate that selecting higher-quality clusters—according to a quality metric we introduce—can decrease the variance penalty without compromising privacy guarantees. Finally, we evaluate the theoretical and empirical performance of our CLUSTER-DP algorithm on both real and simulated data, comparing it to common baselines, including two special cases of our algorithm: its unclustered version and a uniform-prior version.

## 1. Introduction

Measuring causal effects from randomized experiments assumes that participants are willing to share potentially sensitive or private responses to treatment. This assumption is constantly challenged by the rise of privacy concerns and regulations for protecting individuals' online data. Many participants and regulatory guidelines agree with sharing some degree of information, as long as there is plausible deniability, meaning no response can be tracked to any individual. While this can be achieved by sharing only aggregated data, this is often not sufficient to entirely prevent the risk of de-anonymization (Sweeney, 2000; Narayanan & Shmatikov, 2008). Differential privacy is one possible data-sharing framework which diminishes this risk.

Ensuring a differential privacy guarantee often comes at the cost of adding additional noise to the original dataset, which increases the variance of statistical estimators. This poses a challenge for many causal inference applications that aim to obtain the most precise measurements possible of a causal effect. However, not all attributes are sensitive. If non-sensitive attributes can be used to cluster users, we may be able to improve this privacy-variance trade-off.

**Motivating application.** We are specifically motivated by the following real-world scenario: suppose we own an advertising platform, and an advertiser wants to analyze the effect of their advertising on custom user segments (e.g., users who have already signed up for their service versus those who have not). This segmentation is proprietary information that the advertiser does not wish to share with the platform. Meanwhile, the platform mandates that user responses (e.g., whether a user has seen or clicked on an advertisement) can only be shared with the advertiser in a differentially private manner. By obtaining privatized user-level data, the advertiser can analyze the effect along this proprietary dimension. Notably, the platform can share non-private information, such as the user's country or, more granularly, their DMA (designated marketing area), and use this non-private cluster information to improve the privacy-variance trade-off of their differentially-private mechanism. Motivated by this application, *we focus on the problem of releasing privatized user-level data rather than solely protecting the privacy of the final causal effect estimation.* Leveraging the post-processing property of differential privacy (Dwork et al., 2014), this approach provides advertisers with flexibility to conduct further analysis on privatized data without compromising user privacy.

Our paper introduces a novel causal and differentially pri-

[1]Marshall School of Business, University of Southern California, Los Angeles, USA [2]Google Research, New York, USA. Correspondence to: Adel Javanmard <ajavanma@usc.edu>, Jean Pouget-Abadie <jeanpa@google.com>.

*Proceedings of the $42^{nd}$ International Conference on Machine Learning*, Vancouver, Canada. PMLR 267, 2025. Copyright 2025 by the author(s).

vate mechanism CLUSTER-DP that theoretically and empirically improves on the privacy-variance trade-off of other baseline mechanisms by leveraging non-private cluster information about the dataset. To do so, we analyze the privacy-variance trade-off for an intuitive set of causal and differentially private baselines. These baselines assume the existence of a central unit that observes all outcomes, and *computes and shares a privatized dataset* on which causal inference analyses can be run by a third-party. In particular, we show that our mechanism, and its analysis, generalizes baselines like UNIFORM-PRIOR-DP, which samples responses at random from the space of possible outcomes, and CLUSTER-FREE DP, which is a special case when all units belong to the same cluster. As this last baseline indicates, our mechanism imposes no restrictions on the properties of the clusters, except that they each be of cardinality at least two. Our mechanism yields the largest improvements when the clusters exhibit a specific measure of cluster quality.

**Organization of the paper.** In Section 2, we define and motivate the differential privacy setting and causal objective of our work, and discuss several intuitive mechanisms for privatizing a dataset while still allowing for unbiased and consistent estimation of the average treatment effect. In Section 3, we introduce our novel private-and-causal CLUSTER-DP mechanism, and its special case when all units belong to the same cluster, the CLUSTER-FREE DP mechanism. We evaluate their privacy guarantees and their variance gap to their non-differentially-private counterparts. We conclude in Section 4 with numerical experiments on simulated and real graphs to validate our claims and compare the empirical performance of each algorithm.

## 1.1. Related works

There is a vast literature on differential privacy and causal inference. Here, we will discuss only the works closely related to ours to better position our research.

**Causal inference under privacy constraints.** Within the growing body of work recently at the intersection of privacy and causal inference, Kancharla & Kang (2021) explore the problem of estimating average treatment effects in randomized control trials where binary outcomes are privatized using a differentially private mechanism. Their mechanism randomly returns altered responses or the true responses, with some predefined probabilities. In contrast, our work extends beyond binary outcomes to a general discrete outcome space and improves the privacy-variance tradeoff by leveraging a clustering structure of responses, which is quantified by cluster quality. Unlike their approach, which does not account for the empirical distribution of responses, our method offers broader applicability and variance reduction through clustering. In the absence of non-compliers, the procedure in (Kancharla & Kang, 2021) can be viewed as a special case of this work when considering binary outcomes

without clustering (cf. Equation (9)).

Among other works in this space, Guha & Reiter (2024) develops differentially private (DP) algorithms for estimating weighted average treatment effects with binary outcomes by first characterizing the global sensitivities of point and variance estimators. A subsample-and-aggregate approach generates noisy versions of these estimators, followed by a Bayesian procedure to produce interval estimates. In contrast, Lee et al. (2019) use DP empirical risk minimization to estimate propensity scores via a logistic regression model, adding privacy-preserving noise using the Gaussian mechanism. Both (Guha & Reiter, 2024; Lee et al., 2019) operate within the central DP framework, where a trusted curator collects raw data. Ohnishi & Awan (2023) take a local DP approach, where individuals privatize their data before sharing, and propose a minimax optimal estimator for population average treatment effects (ATE) under fixed assignment probabilities, achieving optimal mean-squared error (MSE) rates. The proposed estimator is a special case of Algorithm 3 (with one cluster), a benchmark that we discuss in Section A.4 and in our numerical experiments in Section A.8.

Other works have explored the privacy-utility trade-off in various causal inference settings. Betlei et al. (2021) developed a differentially private method called ADUM for learning uplift models in randomized control trials. Their approach involves adding Laplace noise to aggregated responses within feature-based bins. They examine the privacy-utility trade-off through the mean-squared error of the estimator, although they do not utilize clustering to reduce estimator variance. Niu et al. (2022) introduced a multi-stage learning meta-algorithm under privacy constraints, employing DP-EBMs and sample-splitting to estimate conditional average treatment effects (CATE). Their work focuses on privacy-accuracy trade-offs but does not provide private unit-level data or use clustering techniques to improve variance. These studies underscore ongoing efforts to balance privacy and utility in causal inference, each employing distinct approaches with their own limitations.

**Label-Differential privacy.** The notion of differential privacy measure (Dwork et al., 2006a;b) is widely used in practice and is extensively covered in the literature. In this work, we focus primarily on label differential privacy, introduced by Chaudhuri & Hsu (2011); cf our privacy setting and Remark 2.1. The literature on label differential privacy is mostly dedicated to classification and regression tasks with the goal of improving excess risk while offering protection for labels (Beimel et al., 2013; Bassily et al., 2018; Wang & Xu, 2019).

Our approach draws inspiration from the work of Esfandiari et al. (2022), adapting their technique to the context of causal effect estimation. While their method was devel-

oped for a different purpose, namely reducing excess risk of a model trained on private labels, we have modified and extended it to address the specific challenges of privacy-preserving causal inference. To highlight our contributions, first note that the two works operate under different settings and assumptions: Esfandiari et al. (2022) posits an underlying data-generating law $p(y|x)$ and a cluster heterogeneity measure based on the distance of the "conditional" label distribution of a sample to its cluster, averaged on the entire population. In our work, the features are only used to form the cluster, and nothing beyond that. This allows us to have the flexibility of making our proposal fully DP (using DP clustering per Remark 2.1) and in general better control the privacy leakage of features (if any) as it is only used to make clusters. Additionally, and unlike (Esfandiari et al., 2022), we do not assume any underlying data generative model and the cluster homogeneity we define is simpler, as it is only based on the outcomes and clusters, without conditioning on the features. Our variance analysis contains completely new ideas with no resemblance to the analysis of (Esfandiari et al., 2022). Here, we need to analyze the variance of the ATE and its inflation due to the DP mechanism, which is done in multiple steps, considering the randomness with respect to the treatments, and the DP mechanism separately. In addition, we provide a tighter analysis of the privacy guarantee than the proof methodology of (Esfandiari et al., 2022), and also extend it to an $(\varepsilon, \delta)$-type guarantee (See Theorem 3.1 and Definition 2.2).

## 2. Causal Objective, Privacy Setting, and Common Baselines

We are motivated by a real-world scenario of a technology company selling advertising space to advertisers. Its clients, the advertisers, wish to measure the effectiveness of their advertising campaigns by running A/B tests, but do not want to rely on this technology company to provide their causal estimates. Instead, *they would like access to user-level data*, such as whether a user clicked on their ad, as well as any meaningful covariates about that user. One reason for this might be that advertisers wish to run their own covariate-adjustment methods, or they would like to investigate proprietary sub-slices of users. On the other hand, this technology company seeks to protect the privacy of its users. Hence it must act as a central unit which privatizes its datasets before passing them on to advertisers for them to perform their own causal inference analyses. We now introduce the formal causal objective and privacy setting, as well as a set of common baselines. To guide the reader through abundant notation, we include a glossary at the beginning of the Appendix.

**Causal Objective.** We consider a fixed population of $n$ units where we can assume the Stable Unit Treatment Value Assumption (Imbens & Rubin, 2015). Let $y_i(0)$ be the potential outcome of unit $i$ if it is controlled, and $y_i(1)$ if it is treated, each sampled from a finite response space $\mathcal{Y}$ of cardinality $K = |\mathcal{Y}|$. While finite response spaces are common in many advertising settings (e.g. number of clicks or impressions), we suggest binning when outcomes are continuous, as illustrated later in Section 4.

For our CLUSTER-DP algorithm, we will further assume that there is some known clustered structure of these units into $C = |\mathcal{C}|$ non-overlapping clusters of size $n_c$, and let $c_i \in \mathcal{C}$ be the cluster membership of unit $i$. These clusters can be geographic regions or broad demographic groups the units belong to. We do not make any assumptions on the number of clusters or their size, except that they must contain at least two units. In particular, our results hold for a single large cluster. We show later in Section 3 that our results improve with a specific measure of cluster quality.

Our causal estimand is the average treatment effect estimand defined in the finite sample regime by $\tau = \frac{1}{n} \sum_{i=1}^{n} (y_i(1) - y_i(0))$. While we focus on the common finite sample setting, our results can be extended to their super-population equivalents, in which case, we denote by $x_i \in \mathbb{R}^d$ the covariate vector of each unit $i$ such that each $(x_i, y_i(0), y_i(1), c_i)$ is drawn from some joint distribution $\mathcal{P}$.

Let $z_i$ correspond to the treatment assignment of unit $i$, $z_i = 1$ if treated and $z_i = 0$ if controlled. Let $n_1$ be the total number of treated units and $n_0$ to be the total number of controlled units across units. Treatment and control are assigned completely at random. When a clustering of the data is available, the treatment assignment is sampled in a completely randomized way over clusters: a fixed number of $n_{1,c}$ (resp $n_{0,c}$) units is chosen uniformly at random to be treated (resp. controlled) within each cluster $\mathcal{C}$, with $n_c = n_{1,c} + n_{0,c}$ the total number of units in cluster $c$.

**Privacy Setting.** In the real-world advertising setting presented above, it is assumed that a central unit privatizes the dataset before sharing it externally. Some of the mechanisms we explore also work without the presence of this central unit, a setting known as local differential privacy, but this is not a requirement.

In our setting, we have three key variables: outcomes, treatment assignments, cluster membership. An important question is which of these variables should be privatized. In our advertising setting, a unit's treatment assignment is assigned purely at random; it is therefore not sensitive, and can be shared as-is. Outcomes are clearly sensitive and should be privatized. In our exposition, we primarily assume that the cluster structure is non-sensitive information (as is the case in our motivating application, where clusters can be formed based on user's country of designated market area). Nonetheless, as we discuss in the following remark, we can easily extend it to applications where cluster structure is

also deemed sensitive and should be privatized.

**Remark 2.1.** *The privacy protection of cluster structure can be decoupled from that of outcomes. In particular, by virtue of the composition property of differential privacy (Dwork et al., 2014), we can allocate $\varepsilon_1$ privacy budget for privatizing the clusters and $\varepsilon_2$ privacy budget for privatizing the outcomes. The end-to-end process will be $\varepsilon(=\varepsilon_1+\varepsilon_2)$-DP. It is worth noting that there already exists a rich literature on DP-clustering algorithms, see e.g., (Nissim et al., 2007; Feldman et al., 2009; 2017; Stemmer & Kaplan, 2018; Cohen et al., 2021).*

Following Remark 2.1, we focus on privatizing the outcomes, assuming a given *non-sensitive* cluster structure. This approach aligns with the concept of label-differential privacy, introduced by Chaudhuri & Hsu (2011).

**Definition 2.2.** *(Label Differential Privacy) Consider a randomized mechanism $M : D \rightarrow \mathcal{O}$ that takes as input a dataset $D$ and outputs into $\mathcal{O}$. Let $\varepsilon, \delta \in \mathbb{R}_{\geq 0}$. A mechanism $M$ is called $(\varepsilon, \delta)$-label differentially private— or $(\varepsilon, \delta)$-label DP—if for any two datasets $(D, D')$ that differ in the label (outcome) of a single example and any subset $O \subseteq \mathcal{O}$ we have $\mathbb{P}[M(D) \in O] \leq e^\varepsilon \mathbb{P}[M(D') \in O] + \delta$, where $\varepsilon$ is the privacy budget and $\delta$ is the failure probability. If $\delta = 0$, then $M$ is said to be $\varepsilon$-label differentially private, or $\varepsilon$-label DP.*

The $(\varepsilon, \delta)$-differential privacy property states that it is unlikely that the observed output has a much higher or lower chance to be generated under a dataset $D$ compared to a neighboring dataset $D'$. In our context, a unit's "label" refers to its observed outcome; we use the words outcome and label interchangeably.

It is worth noting that the differential privacy focuses on the privacy loss to an individual *by her contribution to a dataset*. In the context of label-DP, one may wonder that the potential correlation between non-sensitive features and the labels may violate privacy of individuals. In such cases, it is the correlational pattern and the *conclusions reached* that affect the individual, *not her presence or absence in the data set*. We refer to (Dwork et al., 2014)(Chapter 1) for an elaborate discussion.

**Two aggregation-based baselines.** Perhaps the simplest approach to sharing a differentially private estimate of the average treatment effect is for the central unit to compute some unbiased estimator based on the original responses $y_i$ and add noise to the estimate before sharing it externally. We provide in Appendix A.4 a formal description of this approach for the Horvitz-Thompson estimator, defined in Appendix A.2, which we refer to as the NOISY HORVITZ-THOMPSON mechanism, along with a proof of its differential privacy guarantee and its variance gap.

A second and slightly more sophisticated approach would be

for the central unit to add noise to the frequency of responses in each cluster before sharing the histogram externally, since the estimated treatment effect depends only on the histogram of responses of treated and controlled units in each cluster. We refer to this approach as the NOISY HISTOGRAM approach. We provide in Appendix A.4 a formal description of this approach, written in the broadest generality when a clustering is available, as well as a proof of its differential privacy guarantee along with its variance gap.

**Limitations.** These aggregation-based approaches have two drawbacks in the real-world setting described in Section 2. First, in practice, advertisers expect *user-level data, even if privatized*. This is likely because they wish to analyze their own segments of the user population or apply their own proprietary covariate-adjustment methods. Second, the noise of these aggregated approaches is averaged over the number of clusters $C$ (if any) or over the number of possible outcomes $K$. User-level mechanisms on the other hand, add noise that is averaged over $n$, the number of users. This becomes yet another competitive advantage of user-level methods in the case of one-shot communication between the central unit and the advertisers. In particular, a user-level privatizing scheme achieves lower finite-sample conditional bias than the prior two aggregation baselines when $n \gg K$ and $n \gg C$, when we condition on the randomness of the DP mechanism and consider the bias with respect to the randomization in the sub-population. We will illustrate this point further through experiments in Appendix A.8.

**The UNIFORM-PRIOR-DP baseline.** We also consider a differentially private causal mechanism that reports the true outcome with some probability, and otherwise reports an outcome sampled uniformly at random from the space of possible outcomes. We refer to it as the UNIFORM-PRIOR-DP mechanism, formalized in Algorithm 1, because it does not leverage any information about the empirical distribution of outcomes beyond its support. The UNIFORM-PRIOR-DP mechanism is a generalization of the mechanism proposed by (Kancharla & Kang, 2021) in the binary-outcome setting, when there are no non-compliers.

---

**Algorithm 1** UNIFORM-PRIOR-DP mechanism

**Input**: Individual responses $y_1, \ldots, y_n$
**Output**: Privatized responses $\tilde{y}_1, \ldots, \tilde{y}_n$
**for** $i \in \{1, \ldots n\}$ **do**
$$\tilde{y}_i \leftarrow \begin{cases} y_i^0 \sim \mathcal{U}(\mathcal{Y}) & \textit{with probability } \lambda \\ & \textit{// } \mathcal{U} \textit{ is the uniform distribution} \\ y_i & \textit{with probability } 1 - \lambda \end{cases}$$
*Return privatized responses* $\{\tilde{y}_1, \ldots, \tilde{y}_n\}$.

---

In the broadest generality when a clustering is available, the following stratified estimator is unbiased for the average

treatment effect:

$$\hat{\tau}_\lambda = \frac{1}{1-\lambda} \sum_{c \in \mathcal{C}} \frac{n_c}{n} \sum_{i \in c} \left( \frac{\tilde{y}_i z_i}{n_{1,c}} - \frac{\tilde{y}_i(1-z_i)}{n_{0,c}} \right) . \quad (1)$$

As we discuss in Section 3, the UNIFORM-PRIOR-DP mechanism is a special case of our proposed algorithm. We include its privacy guarantees and the variance gap of $\hat{\tau}_\lambda$ in Appendix A.5.

# 3. The CLUSTER-DP and CLUSTER-FREE-DP Mechanisms

We now introduce our differentially private mechanism, CLUSTER-DP, which not only provides user-level privatized outcomes, but also leverages information about the empirical distribution of outcomes within each cluster to improve its variance gap. When no good clustering is available, we consider its special case when all units can be considered part of the same cluster, the CLUSTER-FREE-DP mechanism.

---

**Algorithm 2** Our CLUSTER-DP mechanism

---

**Parameters**: threshold $\gamma \in [0, 1/K]$; noise scale $\sigma \geq 0$; re-sampling probability $\lambda \in [0, 1]$
**Input**: Individual responses $y_1, \ldots, y_n$, treatment assignments $z_1, \ldots, z_n$.
**Output**: Privatized responses $\tilde{y}_1, \ldots, \tilde{y}_n$
**for** $c \in \mathcal{C}$ **do**
  **for** $a \in \{0, 1\}, y \in \mathcal{Y}$ **do**
    $q_a(y|c) \leftarrow \max\{\gamma, \min\{1, \hat{p}_a(y|c)+w\}\}$, where
    $w \sim \text{Laplace}(\sigma/n_{a,c})$
  **for** $a \in \{0, 1\}$ **do**
    **for** $y \in \mathcal{Y}$ **do**
      **if** $\sum_y q_a(y|c) > 1$ **then**
        $\zeta_y \leftarrow q_a(y|c) - \gamma$
      **else**
        $\zeta_y \leftarrow 1 - q_a(y|c)$
    **for** $y \in \mathcal{Y}$ **do**
      $\tilde{q}_a(y|c) \leftarrow q_a(y|c) + \frac{\zeta_y}{\sum_{y'} \zeta_{y'}} \left( 1 - \sum_y q_a(y|c) \right)$
**for** $i \in \{1, \ldots n\}$ **do**
  $\tilde{y}_i \leftarrow \begin{cases} y_i^0 \sim \tilde{q}_{z_i}(\cdot|c_i) & \text{with probability } \lambda \\ y_i & \text{with probability } 1 - \lambda \end{cases}$
*Return privatized responses* $\{\tilde{y}_1, \ldots, \tilde{y}_n\}$.

---

Formalized in Algorithm 2, our proposed mechanism deals with each cluster individually and independently of other clusters, handling treated and controlled groups separately. It returns a privatized potential outcome $\tilde{y}_i$ for each unit, which is either the true outcome with some probability or sampled from a transformed empirical distribution of responses from units in the same cluster. The transformation is inspired from a mechanism in (Esfandiari et al., 2022).

For the sake of exposition, we focus on the controlled units of a cluster $c \in \mathcal{C}$.

(1) Compute the empirical response distribution of the controlled units in the cluster $\hat{p}_0(y|c)$.

(2) Add noise drawn from a Laplace distribution with parameter $(\sigma/n_{0,c})$ to each response probability.

(3) Truncate the response probabilities to be within the interval $[\gamma, 1]$, with $\gamma \leq 1/K$.

(4) Renormalize the response probabilities to form a distribution, i.e. such that the resulting response probabilities remain in $[\gamma, 1]$, and add up to one.

(5) With probability $\lambda$, each original response is replaced by a random sample from this distribution constructed in the previous step.

## 3.1. Privacy guarantees

The following theorem and its corollary state the differential privacy guarantee of our CLUSTER-DP mechanism.

**Theorem 3.1.** *Let $\tilde{\varepsilon} > 0$ and $\delta := \max(0, 1 - \lambda + \lambda\gamma(1 - e^{\tilde{\varepsilon}}))$. The* CLUSTER-DP *mechanism described in Algorithm 2 is $(\varepsilon, \delta)$-label DP with $\varepsilon = \min\left(\frac{1}{\sigma}, \frac{2}{\gamma}\right) + \tilde{\varepsilon}$. By setting $\tilde{\varepsilon} = \log(1 + \frac{1-\lambda}{\lambda\gamma})$, we have $\delta = 0$, and therefore the* CLUSTER-DP *mechanism is also $\varepsilon$-label DP, with $\varepsilon = \min\left(\frac{1}{\sigma}, \frac{2}{\gamma}\right) + \log\left(1 + \frac{1-\lambda}{\lambda\gamma}\right)$.*

We refer the reader to Appendix A.13 for a full proof of Theorem 3.1 and provide here some intuition for its stated privacy loss $\varepsilon$. The first term $\min(1/\sigma, 2/\gamma)$ is the privacy budget used to privately estimate the empirical response distribution $\tilde{q}_a(\cdot|c)$ for each cluster. Fixing the transformed empirical distributions $\tilde{q}_a(\cdot|c)$, the log term is the privacy budget used to generate the privatized responses $\tilde{y}_i$. By the composition theorem for differential privacy (Dwork et al. (2014), Theorem B.1), the total privacy loss is given by the sum of these two losses. As expected, when the resampling probability goes to zero ($\lambda \to 0$), the privacy loss grows large ($\varepsilon \to +\infty$). Similarly, as the Laplace noise $\sigma$ and truncation parameter $\gamma$ grow large, the privacy guarantee improves ($\varepsilon \to 0$).

**The CLUSTER-FREE mechanism.** Because these privacy guarantees do not depend on the size, cardinality, or quality of the clusters, Theorem 3.1 also holds for the special case where there is no cluster structure to the data, in which case we can repeat the same mechanism as if all units belong to the same large cluster. We refer to this mechanism as the CLUSTER-FREE-DP mechanism, it has the same privacy guarantee as the CLUSTER-DP mechanism. We will show the benefit that clusters may have in Section 3.2.

**Similarity with UNIFORM-PRIOR-DP.** The distributions $\tilde{q}_a(y|c)$ constructed in the CLUSTER-DP mechanism obey the following properties: $\tilde{q}_a(y|c) \geq \gamma$ for all $y \in \mathcal{Y}$, and $\sum_y \tilde{q}_a(y|c) = 1$. When setting the truncation parameter $\gamma = 1/K$, these distributions reduce to uniform distributions over the space of all outcomes, in which case the cluster-DP mechanism amounts to the simpler UNIFORM-PRIOR-DP mechanism introduced briefly at the end of Section 2, regardless of the value of the Laplace noise variance $\sigma^2$. We obtain the privacy guarantees of this baseline, detailed in Section A.5, as a corollary of Theorem 3.1.

### 3.2. Estimation and variance guarantees

We now consider estimating causal effects from the privatized outcomes provided by our CLUSTER-DP mechanism. For each cluster $c \in \mathcal{C}$ and each value $a \in \{0, 1\}$ of treatment, we construct the response randomization matrix $Q_{c,a} \in \mathbb{R}^{K \times K}$:

$$Q_{c,a}[y', y] := (1 - \lambda)\mathbb{I}(y' = y) + \lambda \tilde{q}_a(y'|c). \quad (2)$$

Conditional on its true outcome $y_i$, treatment assignment $z_i$, and cluster assignment $c_i$, the privatized response $\tilde{y}_i$ of unit $i$ is distributed according to $Q_{c_i, z_i}[\tilde{y}_i, y_i]$: $\forall y'$, $P(\tilde{y}_i = y'|c_i, z_i, y_i) = Q_{c_i, z_i}[y', y_i]$.

We use the inverse of the response randomization matrix to debias the privatized responses. We use the notation y to represent in vector form the space of all possible potential outcomes, with similar ordering of rows and columns as $Q_{c_i, z_i}$. With a small abuse of notation, we write the index $\tilde{y}_i$ of the vector $\text{y}^T Q_{c, z_i}^{-1}$ as $\text{y}^T Q_{c, z_i}^{-1}[\tilde{y}_i]$ and show that it is an unbiased estimate for $y_i$ over the randomness of Algorithm 2. As a result, by reweighting each privatized outcome by the inverse of its conditional probability of occurring $Q_{c_i, z_i}[\tilde{y}_i, y_i]$, we construct an unbiased and consistent estimator for the average treatment effect in the following Theorem, a proof of which can be found in Appendix A.14.

**Theorem 3.2.** *Consider the following $\hat{\tau}_Q$ estimator,*

$$\hat{\tau}_Q := \sum_{c \in \mathcal{C}} \frac{n_c}{n} \sum_{i \in c} \left( \text{y}^T Q_{c, z_i}^{-1}[\tilde{y}_i] \frac{z_i}{n_{1,c}} - \text{y}^T Q_{c, z_i}^{-1}[\tilde{y}_i] \frac{1 - z_i}{n_{0,c}} \right)$$

*Conditionally on the randomness of the treatment assignment, $\hat{\tau}_Q$ is equal in expectation over the randomness of the DP mechanism to the stratified difference-in-means estimator below, henceforth $\hat{\tau}_{\text{NO-DP}}$, which also implies that $\hat{\tau}_Q$ is an unbiased and consistent estimator of $\tau$.*

$$\mathbb{E}_{DP}[\hat{\tau}_Q|\boldsymbol{z}] = \sum_{c \in \mathcal{C}} \frac{n_c}{n} \left( \sum_{i=1}^{n} y_i(1) \frac{z_i}{n_{1,c}} - \sum_{i=1}^{n} y_i(0) \frac{1 - z_i}{n_{0,c}} \right)$$

For third parties to compute this estimator themselves, the central unit must pass along the cluster assignment, the treatment assignment, the privatized response $\tilde{y}_i$, as well as the vector of probabilities $\text{y}^T Q_{c, z_i}^{-1}$. An illustration is included in Figure 1.

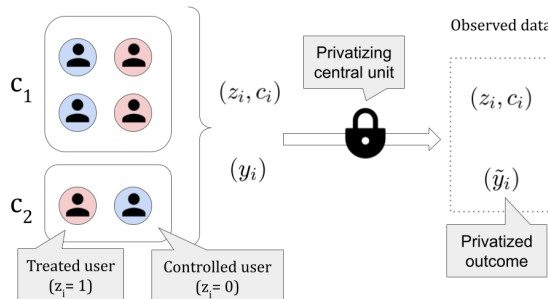

Figure 1: Illustration of CLUSTER-DP mechanism with a central unit computing the (clustered) privatized outcomes for valid causal inference.

Since $\tilde{q}_a(\cdot|c)$ and the responses $\tilde{y}_i$ are $\varepsilon$-DP, by the post-processing property of differential privacy (Dwork et al. (2014), Proposition 2.1), all the information passed to the third-party, as well as any estimation based on this information, is also $\varepsilon$-DP.

**Remark 3.3.** *Note that the estimator $\hat{\tau}_Q$ depends on $Q_{c,a}^{-1}$ for $a \in \{0, 1\}$. From (2), $Q_{c,a}$ is a scalar of the identity matrix plus a rank one matrix. As shown in the proof of Proposition A.9 (cf. (27)), $Q_{c,a}^{-1}$ is also a rank one perturbation of the (scaled) identity matrix. In Lemma A.11 we bound the maximum singular value of $Q_{c,a}^{-1}$. Using this lemma, the minimum singular value of $Q$ is at least $(1-\lambda)/(\lambda\sqrt{K}+1)$. Therefore, $Q$ is well-conditioned if $\lambda < 1$, which will be the case by choosing $\sigma < \varepsilon$ in our privacy bound in Theorem 3.1. This is also reflected in the variance bound, Theorem 3.5, where $(1 - \lambda)^2$ appears in the denominator.*

Our goal for Algorithm 2 is to make the gap between the variance of our differentially-private estimator $\hat{\tau}$ and its non-differentially private counterpart $\hat{\tau}_{\text{NO-DP}}$ as small as possible for a given privacy guarantee. While all our results hold for any given clustering, they are greatly improved when clusters are homogeneous.

**Definition 3.4** (Cluster homogeneity). *For $a \in \{0, 1\}$, define a clustering's homogeneity as the average intra-cluster variance of outcomes $\phi_a \geq 0$:*

$$\phi_a := \sum_{c \in \mathcal{C}} \frac{n_c^2}{n^2} \frac{S^2(\vec{y}_c(a))}{n_{a,c}},$$

*where for vector $\vec{u} \in \mathbb{R}^d$, $S^2(\vec{u}) := \frac{1}{d-1} \sum_{u \in \vec{u}} (u - \bar{u})^2$.*

The quantity $\phi_a$ has a natural super-population interpretation when taking its expectation of over the distribution $\mathcal{P}$:

$\phi_a = \mathbb{E}[\text{Var}(y(a)|c)] = \text{Var}(y(a)) - \text{Var}(\mathbb{E}[y(a)|c]) > 0$. Holding $\text{Var}(y(a))$ constant, lower values of $\phi_a$ implies that clusters are better separated. For $\phi_a = 0$, the outcome values of each clusters are contained within a singleton set. On the other hand, if $\phi_a$ is high, clusters contain a wide range of responses, up to the variation of outcomes of the population.

We now state our key theorem, which provides a bound on the variance of our estimator $\hat{\tau}_Q$, with respect to the randomness of Algorithm 2 and the random assignment $z$, as a function of $\text{Var}_z[\hat{\tau}_{\text{No-DP}}]$ and $\phi_a$.

**Theorem 3.5.** *The variance of $\hat{\tau}_Q$ defined in Theorem A.14 is bounded by*

$$0 \le \text{Var}_{DP,z}[\hat{\tau}_Q] - \text{Var}_z[\hat{\tau}_{\text{No-DP}}]$$

$$\le \left( \frac{1}{(1-\lambda)^2} - 1 \right) \sum_{a \in \{0,1\}} \phi_a + \sum_{a \in \{0,1\}} \sum_{c \in \mathcal{C}} \frac{n_c^2}{n^2} \frac{A(n_{a,c})}{n_{a,c}},$$

*where $\phi_a$ is the measure of cluster homogeneity defined in Definition 3.4, and for any $x$,*

$$A(x) := 2K \left[ \gamma + \frac{\sigma}{x} \left( e^{-\gamma x/\sigma} - e^{-x/\sigma} \right) \right] \left[ 2\|\mathbf{y}\|_\infty^2 + \dots \right.$$

$$\left. \dots \frac{3\|\mathbf{y}\|_\infty^2 + (\lambda\sqrt{K}+1)^2 + \|\mathbf{y}\|_2^2(1-\lambda(K-1)\gamma)}{(1-\lambda)^2} \right],$$

*with $K$ the number of possible outcomes and $\mathbf{y} \in \{\mathcal{Y}\}^K$ the vector of all possible outcomes.*

Theorem 3.1 and Theorem 3.5 together allow us to capture the privacy-variance trade-off of our proposed mechanism. Recall that the privacy guarantee of Theorem 3.1 is agnostic to the clustering. On the other hand, the variance gap in Theorem 3.5 depends first on the homogeneity of clusters, as defined in Definition 3.4, and a second term that is agnostic to the clustering. As a result, more homogeneous clusters—those with low $\phi_a$—result in a smaller variance gap with equal privacy guarantees, leading to a better privacy-variance trade-off than less homogeneous clusters.

We can also provide some intuition for the second term $A(x)$. By choosing $\gamma$ and $\sigma$ to be arbitrarily small, we can make this second term arbitrarily small. As expected, the privacy guarantees of Theorem 3.1 suffer in that regime. When setting $\lambda = 0$, our CLUSTER-DP mechanism always outputs the true outcome, and we no longer produce privatized outcomes. In that case, we can set the truncation parameter $\gamma$ and the Laplace noise $\sigma$ to be zero with no consequence to recover the trivial equality $\text{Var}_{DP,z}(\hat{\tau}) = \text{Var}_z[\hat{\tau}_{\text{No-DP}}]$ from our bound above. Naturally, the more interesting setting from a privacy perspective is $\lambda \in (0,1)$.

As discussed previously, the privacy guarantee in Theorem 3.1 for the CLUSTER-DP and CLUSTER FREE-DP

mechanisms reduces to the guarantee of the UNIFORM-PRIOR-DP mechanism in Theorem A.5 when setting the truncation parameter $\gamma = 1/K$ and $\sigma = \infty$. Because both CLUSTER-DP and CLUSTER-FREE-DP mechanisms use data-dependent priors, there may exist choices of $(\sigma, \gamma, \lambda)$ which result in better privacy-variance trade-offs than the latter for certain outcome distributions. In the following section, we conduct empirical evaluations of the privacy-variance trade-off of the different mechanisms.

# 4. Numerical Experiments

In this section, we perform several experiments to validate the claims we make in the paper and to illustrate their usefulness. Due to the strong limitations of the aggregation-based baselines in our real-world setting, we focus our attention on three UNIFORM-PRIOR DP, CLUSTER-FREE-DP, and CLUSTER-DP mechanisms, and relegate further experimental investigations of these aggregation-based baselines to Appendix A.8.

We start by considering a Gaussian Mixture Model setting where for every unit $i$ in cluster $c$, a continuous quantity $y_i'$ is given by

$$\forall i \in c, \, y_i' = \sqrt{\beta}\mu_c + \sqrt{v - \beta}w_i, \tag{3}$$

where $\mu_c$ and $w_i$ are drawn from the standard normal distribution. The coefficient $\beta \in [0, v]$ measures the dependence of the response on the cluster center. This specific parameterization is chosen to fix the variance of the response, equal to $v$, as $\beta$ varies. Since the proposed mechanism is for discrete outcome spaces, we quantize the response in the following way:

$$y_i(1) = y_i(0)+\tau \quad \text{and} \quad y_i(0) = \begin{cases} K' & \text{if } y_i' > 2\sqrt{v} \\ -K' & \text{if } y_i' < -2\sqrt{v} \\ [y/\Delta] & \text{otherwise} \end{cases}$$

where $\Delta := 2\sqrt{v}/K'$ and $[x]$ denotes the rounding of $x$ to the nearest integer. The treatment effect is an additive $\tau$ term on the potential outcome under control. We fix $\tau = 1$, such that the outcomes take values in the set $\mathcal{Y} = \{-K', \dots, 0, \dots, K', K'+1\}$. We denote by $K := 2(K'+1)$ the size of outcome space.

Unless otherwise specified, and with no particular reason to fix parameters one way or another, we take $K' = 5$, $v = 5$, and $\beta = 4.5$. We consider $C = 3$ clusters of sizes $500, 10^3, 2 \times 10^3$ with an equal number of controlled and treated units in each cluster. To display confidence intervals around certain results, we consider a super-population of three clusters of sizes $2.5 \times 10^3$, $5 \times 10^3$, and $10^4$ units, and repeatedly draw uniformly at random sub-populations of three clusters from these original clusters.

For any given sub-population, we compute the variance $\text{Var}_{DP,z}[\hat{\tau}_Q]$ by empirically computing the variance (or histogram) of $\hat{\tau}_Q$ empirically over 500 realizations of the randomness in the corresponding DP mechanism (e.g. Laplace noise and response randomization), as well as the treatment assignments, which are done by choosing balanced set of treated and controlled units uniformly at random within each cluster. Unless otherwise specified, for CLUSTER-DP mechanism, we set the truncation parameter $\gamma = 0.02$, the Laplace noise $\sigma = 10$, and the resampling probability $\lambda = 0.8$.

**Experiment 1. (Privacy-variance trade-off)** We compare the privacy-variance trade-off of our suggested CLUSTER-DP mechanism with the CLUSTER-FREE-DP mechanism, as well as the stratified and unstratified versions of the UNIFORM-PRIOR-DP mechanism. We observe that the CLUSTER-DP can have significantly lower variance for its estimator, compared to the other mechanisms, for the same privacy loss $(\varepsilon, \delta)$.

In Figure (1.a), we aim to fix the privacy loss to $\varepsilon = 0.2$ and $\delta = 10^{-4}$ for all three mechanisms. For the CLUSTER-DP and CLUSTER-FREE-DP, we set the Laplace parameter to $\sigma = 10$, and vary the truncation parameter $\gamma \in [0.1/K, 1/K]$. Following Theorem 3.1, we first choose $\tilde{\varepsilon}$ so that the corresponding privacy $\varepsilon$, is equal to its target $\varepsilon = 0.2$, and then choose the re-sampling probability $\lambda$ to obtain the failure probability $\delta = 10^{-4}$. Likewise, for the UNIFORM-PRIOR-DP mechanism, we set the re-sampling probability $\lambda$ according to Theorem A.5, such that $\varepsilon = 0.2$ and $\delta = 10^{-4}$. In summary, as the truncation parameter $\gamma$ varies, we compare the three mechanisms at the same privacy loss. As we observe in Figure (1.a), for small values of $\gamma$, the CLUSTER-DP achieves significantly lower variance compared to to the other mechanisms. When $\gamma = 1/K$ and $\sigma = \infty$, the theory tells us that CLUSTER-DP reduces to UNIFORM-PRIOR-DP (stratified) and the CLUSTER FREE-DP reduces to UNIFORM-PRIOR-DP (unstratified). However, since we have set $\sigma = 10$, we observe that the variance for the UNIFORM-PRIOR-DP becomes lower than the other mechanisms for $\gamma = 1/K$. The error-bars here correspond to 50 independent draws of the sub-population.

In Figure (1.b), we plot the variance of each estimator versus its privacy loss $\varepsilon$, as we fix $\delta = 10^{-4}$. Here, we optimize the choice of Laplace parameter $\sigma \in \{10, 20, \infty\}$ and the truncation parameter $\gamma \in \{0.01/K, 0.1/K, 1/K\}$. We observe that both CLUSTER-DP and CLUSTER FREE-DP estimators achieve a better trade-off than either version of the UNIFORM-PRIOR-DP mechanism. Furthermore, the CLUSTER-DP mechanism, which also leverages the clustering structure, showcases an even better trade-off compared to the CLUSTER FREE-DP mechanism.

**Experiment 2. (Role of clustering quality)** In this experiment we show that, as the clustering quality improves, the variance of the estimator for the CLUSTER-DP mechanism decreases when compared to the variance of the estimator for the CLUSTER FREE-DP mechanism, without affecting their privacy guarantees, since these are agnostic to the clustering according to Theorem 3.1. Under our specified potential outcome model (3), the cluster homogeneity $\phi_a$, as defined in Definition 3.4, is given by $\phi_0 = \mathbb{E}(\text{Var}(y_i(0)|c)) \propto v - \beta = \phi_1$, hence our clusters become more homogeneous as $\beta$ increases. From Theorem 3.5, the clustering structure reduces the variance of the estimator at more homogeneous clusters, i.e. lower values of $\phi_0, \phi_1$, and $\lambda$. We verify this in Figure (1.c), which plots the ratio of the variances for two values of $\lambda \in \{0.5, 0.8\}$ as we vary $\beta$. As $\beta$ grows, we observe a stronger reduction in the variance using the clustering structure of data. This effect is stronger at smaller values of $\lambda$.

**Experiment 3. (Simulation on YouTube data)** Finally, we validate our results on a subset of the YouTube social network to replicate two experiment results in a setting with natural clusters. We compare the variance of our suggested estimator for the CLUSTER-DP mechanism with its variance when using the CLUSTER FREE-DP mechanism to show the benefit of leveraging the clustering structure, replicating the results of Experiment 1.

The YouTube social network dataset (Leskovec & Krevl, 2014) contains the friendship links of a set of users on YouTube, and the ground-truth clusters correspond to groups created by users. We form a smaller dataset, by considering only the 50 largest communities, which includes a total of 22,179 users with a minimum cluster size of 199. We generate the potential outcomes for the users as follows:

$$y_i(0) = x_i^{\mathsf{T}}\beta + w_i, \quad y_i(1) = y_i(0) + \tau,$$

with $w_i \sim \mathsf{N}(0, v^2)$ capturing individual $i$'s effect and the $x_i^{\mathsf{T}}\beta$ term capturing the cluster-level effect. We follow a similar model as in (Zhou et al., 2020) and consider a four-dimensional feature vector $x_i$, with $x_{i1}$ being the number of nodes in cluster $c_i$ (the cluster of user $i$), $x_{i2}$ the number of edges in $c_i$, $x_{i3}$ the number of edges in $c_i$ with other clusters, and $x_{i4}$ the density of cluster $c_i$. Recall that for a cluster with $n$ nodes and $e$ edges, its density is defined as $\frac{e}{\binom{n}{2}}$.

Since the proposed mechanism is for discrete outcome spaces, we quantize the responses into $K = 8$ levels. We standardize the features by making each of the four features zero mean and unit norm across clusters, and setting the standard deviation of the Gaussian noise $w_i$ to $v = 0.1$. In our experiments, we set $\beta = (1, 1, 1, 1)^{\mathsf{T}}$ and $\tau = 1$. In the CLUSTER-DP mechanism, we set the truncation threshold to $\gamma = 0.1/K$ and the Laplace noise level to $\sigma = 5$.

In Figure (1.d), we plot the privacy-variance trade-off for the

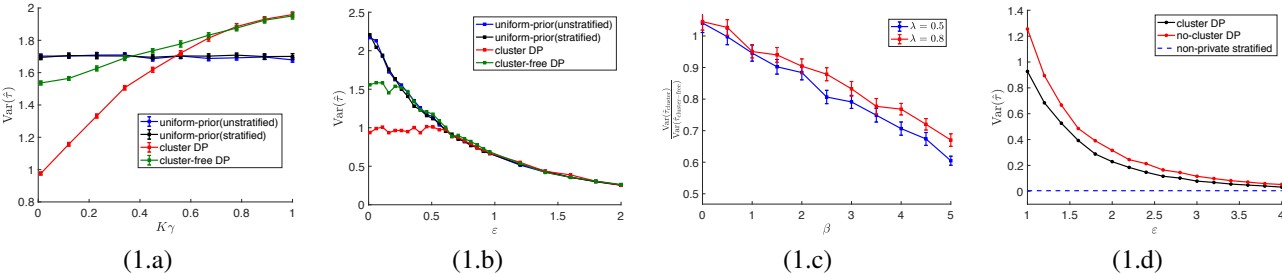

(1.a)  (1.b)  (1.c)  (1.d)

Figure 2: Fuller figures can be found in Appendix A.10. (1.a) Variances of each mechanism as we vary the truncation level $\gamma \in [0.1/K, 1/K]$ in Experiment 1. Privacy loss fixed at $\varepsilon = 0.2$ and $\delta = 10^{-4}$. (1.b) Privacy-variance trade-off of each mechanism under the setting of Experiment 1. We fix the DP failure probability to $\delta = 10^{-4}$, and optimize the choice of $\sigma$ and $\gamma$ in the sets $\sigma \in \{10, 20, \infty\}$ and $\gamma \in \{0.01/K, 0.1/K, 1/K\}$. (1.c) Ratio of the variance of the estimators under the CLUSTER-DP and CLUSTER FREE-DP mechanisms in Experiment 2. The benefit of CLUSTER-DP mechanism is stronger at larger $\beta$ and smaller value of $\lambda$. (1.d) Privacy-variance trade-off of the CLUSTER-DP and CLUSTER FREE-DP stratified estimators for the YouTube dataset in Experiment 3. The dotted line is the variance of the non-private stratified estimator.

CLUSTER-DP and the CLUSTER FREE-DP mechanisms, along with the variance of the non-private stratified estimator, finding once again that the CLUSTER-DP mechanism achieves a better trade-off by leveraging the natural cluster structure of the Youtube users.

**In Appendix.** We include further experimental investigations in Appendix A.6, A.8, and A.9, to verify, amongst others the unbiasedness and consistency of our estimators, as well as illustrate the limitations of aggregation-based baselines against either of these three user-level mechanisms. We also include fuller versions of the subfigures in Figure 2.

## 5. Conclusion

Among differentially private algorithms that allow for valid causal inference, our approach leverages the presence of a non-private clustering structure to minimize the variance gap, as a function of cluster quality, while maintaining privacy guarantees constant. Our procedure generalizes a cluster-free procedure, which we propose and compare to, as well as a more common uniform-prior baseline. We find that, theoretically and empirically on synthetic and semi-synthetic data, our approach outperforms these two methods. Furthermore, our setting, motivated by a real advertising scenario, precludes the use of aggregation-based methods, which we investigate as well.

## Acknowledgment

Adel Javanmard is supported in part by the NSF Award DMS-2311024, the Sloan fellowship in Mathematics, an Adobe Faculty Research Award, an Amazon Faculty Research Award, and an iORB grant from USC Marshall School of Business. The authors are grateful to anonymous reviewers for their feedback on improving this paper.

## Impact Statement

This paper presents work whose goal is to advance the field of Machine Learning. There are many potential societal consequences of our work, none of which we feel must be specifically highlighted here

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

# A. Appendix

## A.1. Notation and common formulas

We recall here all the notations used in the paper and in the proofs:

| | |
|---|---|
| $n$ | total number of units. |
| $n_0$ (resp. $n_1$) | total number of controlled (resp. treated) units. |
| $z_i$ | treatment assignment of unit $i$ in $\{0, 1\}$, with $z_i = 1$ (resp. 0) indicating treatment (resp. control). |
| $\mathcal{Y}$ | response space of cardinality $K = |\mathcal{Y}| < \infty$. |
| $y_i(z)$ | potential outcome in $\mathcal{Y}$ of unit $i$ under treatment assignment $z$. |
| $\tilde{y}_i$ | privatized outcome in $\mathcal{Y}$ of unit $i$ returned by the DP mechanism. |
| y | vector notation of the entire response space $(y)_{y \in \mathcal{Y}}$. |
| $\mathcal{C}$ | set of all clusters of cardinality $C = |\mathcal{C}| < \infty$. |
| $c_i$ | cluster membership of unit $i$. |
| $n_c$ | number of units in cluster $c$, equal to $|\{i \in [n] : c_i = c\}|$. |
| $\mathcal{O}_{a,c}$ | units belonging to cluster $c$ with treatment assignment $a$, equal to $\{i : c_i = c, z_i = a\}$. |
| $\mathcal{O}_c$ | units belonging to cluster $c$, equal to $\{i : c_i = c\}$. |
| $n_{a,c}$ | $|\mathcal{O}_{a,c}|$ for $a \in \{0, 1\}$. |
| $\gamma$ | minimum of clipped empirical distribution, in $[0, 1/K]$. |
| $\sigma$ | noise scale. |
| $1 - \lambda$ | true response sampling probability. |
| $p_a(y|c)$ | true distribution for treated ($a = 1$) and controlled ($a = 0$) units within cluster $c$, equal to $\mathbb{P}(Y(a) = y|c_i = c)$. |
| $\hat{p}_a(y|c)$ | empirical distribution for treated ($a = 1$) and controlled ($a = 0$) units within cluster $c$, equal to $\dfrac{|\{i : c_i = c, z_i = a, y_i = y\}|}{|\{i : c_i = c, z_i = a\}|}$. |
| $\phi_a$ | measure of cluster quality, equal to $\mathrm{Var}(\mathbb{E}[Y(a)|c_X])$. |
| $Q_{c,a}[y', y]$ | response randomization matrix equal to $(1 - \lambda)\mathbb{I}(y' = y) + \lambda\tilde{q}_a(y'|c)$. |
| $u_{a,c}$ | $\sum_{i \in \mathcal{O}_{a,c}} \sum_{y' \in \mathcal{Y}} Q_{c_i,a}^{-1}[y', \tilde{y}_i]y'$. |
| $\vec{y}_{a,c}$ | vector of outcomes observed by the central unit in cluster $c$ with treatment assignment $a$, equal to $\{y_i(a) : i \in \mathcal{O}_{a,c}\}$, for $a \in \{0, 1\}$. |
| $\vec{y}_c(a)$ | all potential outcomes corresponding on the units in cluster $c$ and treatment assignment $a$, equal to $\{y_i(a) : c_i = c\}$. Note that $\vec{y}_c(a)$ contains unobserved values. |
| $e_\ell$ | indicator vector in $\mathbb{R}^{K \times 1}$ with 1 at the $\ell$-th position and zero everywhere else. |
| $A(x)$ | recurring expression in variance bounds, equal to $2K\Big\{B^2\Big(\frac{3}{(1-\lambda)^2} + 2\Big) + \frac{(\lambda\sqrt{K}+1)^2}{(1-\lambda)^2}\|\mathcal{Y}\|^2(1 - \lambda(K-1)\gamma)\Big\}\Big[\gamma + \frac{\sigma}{x}\Big(e^{-\gamma x/\sigma} - e^{-x/\sigma}\Big)\Big]$. |

When the context is clear, we sometimes adopt this lightened notation

| | |
|---|---|
| $\hat{p}_\ell$ | empirical probability of outcome $\ell$ for the controlled units in cluster $c$, equal to $\frac{1}{n_{0,c}}\lvert\{i \in \mathcal{O}_{0,c} : y_i = \ell\}\rvert$. |
| $\hat{\boldsymbol{p}}$ | empirical distribution of outcomes of the controlled units in cluster $c$, arranged into a $K$-dimensional vector, with coordinates $\hat{p}_\ell$, equal to $[\hat{p}_l]_{l\in\mathcal{Y}} \in \mathbb{R}^{K\times 1}$. |
| $Q$ | $Q_{c,0}$ |
| $Q_{a,b}$ | $Q_{c,0}[a,b]$, |
| $Q_{a,b}^{-1}$ | $Q_{c,0}^{-1}[a,b]$, |
| $Q^{-\mathsf{T}}$ | $(Q^{-1})^{\mathsf{T}}$. |
| $\tilde{\boldsymbol{q}}$ | distribution constructed in the DP mechanism (after adding noise to empirical distribution $\hat{p}_{0,D}$, truncation and normalization), equal to $[\tilde{q}_0(y\vert c)]_{y\in\mathcal{Y}} \in \mathbb{R}^{K\times 1}$. |
| $\tilde{q}_l$ | coordinate $l$ of vector $\tilde{\boldsymbol{q}}$. |

## A.2. Variance of the Horvitz-Thompson estimator

The variance of the stratified Horvitz-Thompson estimator is well-known. We recall it below:

$$\hat{\tau}_{\text{No-DP}} := \sum_{c\in\mathcal{C}} \frac{n_c}{n} \sum_{i\in c} \left( \frac{y_i z_i}{n_{1,c}} - \frac{y_i(1-z_i)}{n_{0,c}} \right) .$$

Let $\vec{y}_c := \{y_i : c_i = c\} \in \mathcal{Y}^{n_c}$ be the vector of outcomes of units in cluster $c$, and $\vec{\tau}_c := \vec{y}_c(1) - \vec{y}_c(0)$ be the vector of the differences between each unit's potential outcome in treatment and in control. The variance of $\hat{\tau}_{\text{No-DP}}$ is given by

$$\text{Var}_{\boldsymbol{z}}[\hat{\tau}_{\text{No-DP}}] = \sum_{c\in\mathcal{C}} \frac{n_c^2}{n^2} \left( \frac{S^2(\vec{y}_c(1))}{n_{1,c}} + \frac{S^2(\vec{y}_c(0))}{n_{0,c}} - \frac{S^2(\vec{\tau}_c)}{n_c} \right) ,$$

where, for any vector $u \in \mathbb{R}^d$, $S^2(\vec{u}) := \frac{1}{d-1}\sum_{u\in\vec{u}}(u-\bar{u})^2$ and $\bar{u} := \frac{1}{d}\sum_{u\in\vec{u}}u$. The formula for $\text{Var}_{\boldsymbol{z}}[\hat{\tau}_{\text{No-DP}}^u]$ can be obtained from the formula above where all units belong to a single cluster $\lvert\mathcal{C}\rvert = 1$ and $n_c = n$.

## A.3. Definition of Label Differential Privacy

We recall here a formal definition of label differential privacy.

**Definition A.1.** *(Label Differential Privacy) Consider a randomized mechanism $M : D \to \mathcal{O}$ that takes as input a dataset $D$ and outputs into $\mathcal{O}$. Let $\varepsilon, \delta \in \mathbb{R}_{\geq 0}$. A mechanism $M$ is called $(\varepsilon, \delta)$-label differentially private—or $(\varepsilon, \delta)$-label DP—if for any two datasets $(D, D')$ that differ in the label (outcome) of a single example and any subset $O \subseteq \mathcal{O}$ we have $\mathbb{P}[M(D) \in O] \leq e^\varepsilon \mathbb{P}[M(D') \in O] + \delta$, where $\varepsilon$ is the privacy budget and $\delta$ is the failure probability. If $\delta = 0$, then $M$ is said to be $\varepsilon$-label differentially private, or $\varepsilon$-label DP.*

Achieving label-differential privacy implies that the output of a mechanism does not change too much if a single label in the input dataset is changed. The *privacy loss $\varepsilon$* controls the size of the possible change, and $\delta$ is the *failure probability* in providing such a guarantee. In other words, $(\varepsilon, 0)$-differential privacy ensures that, for *every* run of the mechanism $M$, the observed output is (almost) equally likely to be observed on every other neighboring dataset, simultaneously. The $(\varepsilon, \delta)$-differential privacy property relaxes this constraint and states only that it is unlikely that the observed value $M(D)$ has a much higher or lower chance to be generated under a dataset $D$ compared to a neighboring dataset $D'$. Differential privacy can also be viewed from a statistical hypothesis testing framework, where an attacker aims to distinguish $D$ from $D'$ based on the output of the mechanism. This viewpoint has been put forward by (Wasserman & Zhou, 2010) and (Kairouz et al., 2015), who show that, by using the output of an $(\varepsilon, \delta)$-DP mechanism, the power of any test with significance level $\alpha \in [0, 1]$ is bounded by $e^\varepsilon \alpha + \delta$. For small enough $(\varepsilon, \delta)$, this bound is only slightly larger than $\alpha$, and so any test which aims to distinguishing $D$ from $D'$ is powerless.

## A.4. Two aggregation-based baselines: description and guarantees.

The simplest approach to sharing a differentially private estimate of the average treatment effect is for the central unit to compute some unbiased estimator based on the original responses $y_i$ and add noise to the estimate before sharing it

externally. We provide an example in Algorithm 3, written in the broadest generality when a clustering is available. When no clustering is available, we can simply assume that all units belong to the same cluster.

---

**Algorithm 3** NOISY HORVITZ-THOMPSON mechanism

**Input**: Individual responses $y_1, \ldots, y_n$, (optional) cluster memberships $c_1, \ldots, c_n$
**Output**: Privatized estimate $\hat{\tau}$

$$\text{Return} \quad \hat{\tau} := \sum_{c \in \mathcal{C}} \frac{n_c}{n} \left\{ \sum_{i \in c} \left( \frac{y_i z_i}{n_{1,c}} - \frac{y_i (1 - z_i)}{n_{0,c}} \right) + w_c \right\}, \quad w_c \sim \text{Laplace}(\eta_c). \quad (4)$$

---

The variances of the noise parameters $\eta_c$ determine both the privacy guarantee $\varepsilon$ and additional estimator variance of the Noisy Horvitz-Thompson algorithm. To compute its privacy guarantee, we apply (Dwork et al. (2014), Theorem 3.6) and consider the sensitivity $\Delta_c$ of the inner function $1/n_{1,c} y_i z_i - 1/n_{0,c} y_i (1 - z_i)$, defined as the maximum change in its value when changing only one label in the data set. The variance of $\hat{\tau}$ can be expressed easily as a function of the variance of its non-differentially-private equivalent, the Horvitz-Thompson estimator without the Laplace noise:

$$\hat{\tau}_{\text{No-DP}} := \sum_{c \in \mathcal{C}} \frac{n_c}{n} \sum_{i \in c} \left( \frac{y_i z_i}{n_{1,c}} - \frac{y_i (1 - z_i)}{n_{0,c}} \right). \quad (5)$$

**Proposition A.2.** *The noisy Horvitz-Thompson estimator $\hat{\tau}$ is $\varepsilon$-DP when setting $\eta_c = \Delta_c / \varepsilon$ for every cluster, where $\Delta_c = \min\{n_{0,c}, n_{1,c}\}^{-1} \times \max_{y \in \mathcal{Y}} |y|$. Furthermore, its variance with respect to the treatment assignment $z$ and the Laplace noise $(DP)$ is given by*

$$\text{Var}_{DP,z}[\hat{\tau}] = \text{Var}_z[\hat{\tau}_{\text{No-DP}}] + 2 \sum_{c \in \mathcal{C}} \left( \frac{n_c}{n} \frac{\Delta_c}{\varepsilon} \right)^2,$$

*where $\hat{\tau}_{\text{No-DP}}$ is the non-differentially-private stratified Horvitz-Thompson estimator defined in Eq. 5, and $\text{Var}_z[\hat{\tau}_{\text{No-DP}}]$ is its variance with respect to the treatment assignment $z$.*

Because these results hold for any clustering, they also hold when no clustering is available; in that case, we consider all units to be part of the same cluster. We refer the reader to Appendix A.2 for the well-known closed-form expression of $\text{Var}_z[\hat{\tau}_{\text{No-DP}}]$.

A second and slightly more sophisticated approach would be for the central unit to add noise to the frequency of responses in each cluster before sharing the histogram externally, since the estimated treatment effect depends only on the histogram of responses of treated and controlled units in each cluster. We provide an example in Algorithm 4, written in the broadest generality when a clustering is available.

---

**Algorithm 4** NOISY HISTOGRAM mechanism

**Input**: Individual responses $y_1, \ldots, y_n$, (optional) cluster memberships $c_1, \ldots, c_n$
**Output**: Privatized estimate $\hat{\tau}$
Compute the empirical distribution $\hat{p}_a(y|c)$ of treated ($a = 1$) and controlled ($a = 0$) units within cluster $c$.

$$\text{Return} \quad \hat{\tau} := \sum_{c \in \mathcal{C}} \frac{n_c}{n} \sum_{y \in \mathcal{Y}} y \times (\hat{p}_1(y|c) + w_{1,c,y} - \hat{p}_0(y|c) - w_{0,c,y}), \quad w_{a,c,y} \sim \text{Laplace}(\eta_{a,c}). \quad (6)$$

---

Since the $K$ bins corresponding to the $K$ elements of $\mathcal{Y}$ are disjoint, and the sensitivity of the value of each histogram bin is $n_{a,c}^{-1}$, the central unit can share the histogram privately by adding independent draws from $\text{Laplace}((n_{a,c}\varepsilon)^{-1})$ to the frequency of each value. Furthermore, we can compute in closed form the variance gap of the Noisy Histogram mechanism compared to its non-private Horvitz-Thompson counterpart.

**Proposition A.3.** *The noisy Histogram mechanism $\hat{\tau}$ is $\varepsilon$-DP when setting $\eta_{a,c} = (n_{a,c}\varepsilon)^{-1}$ for every cluster. Furthermore, its variance with respect to the treatment assignment $z$ and the Laplace noise $(DP)$ is given by*

$$\mathrm{Var}_{DP,\boldsymbol{z}}[\hat{\tau}] = \mathrm{Var}_{\boldsymbol{z}}[\hat{\tau}_{\text{No-DP}}] + \frac{2}{\varepsilon^2}\left(\sum_{y\in\mathcal{Y}} y^2\right)\sum_{c\in\mathcal{C}}\left(\frac{n_c}{n}\right)^2\left(\frac{1}{n_{0,c}^2} + \frac{1}{n_{1,c}^2}\right),$$

*where $\hat{\tau}_{\text{No-DP}}$ is the non-differentially-private stratified Horvitz-Thompson estimator defined in Eq. 5, and $\mathrm{Var}_{\boldsymbol{z}}[\hat{\tau}_{\text{No-DP}}]$ is its variance with respect to the treatment assignment $\boldsymbol{z}$.*

For the same privacy guarantee, the NOISY-HORVITZ-THOMPSON mechanism has a smaller variance gap than the NOISY-HISTOGRAM mechanism, since $\|y\|_\infty \le \|y\|_2$ and $\min(n_{0,c}, n_{1,c})^{-2} \le \min(n_{0,c}, n_{1,c})^{-2} + \max(n_{0,c}, n_{1,c})^{-2} = n_{0,c}^{-2} + n_{1,c}^{-2}$.

A proof of Propositions A.3 and A.2 can be found below in Appendix A.11.

### A.5. The UNIFORM-PRIOR DP mechanism

Unlike the two prior mechanisms, the UNIFORM-PRIOR DP mechanism provides user-level outcomes. As formalized in Algorithm 1, it reports the true outcome with some probability, and otherwise reports an outcome sampled uniformly at random from the space of possible outcomes.

Our next result shows that the stratified estimator (1) is unbiased for the average treatment effect.

**Proposition A.4.** *The conditional expectation of the estimator $\hat{\tau}_\lambda$ defined in Eq. (1), with respect to the DP mechanism, is equal to the non-differentially private Horvitz-Thompson estimator. It is therefore unbiased for the average treatment effect $\tau$ over $\boldsymbol{z}$ and the DP mechanism.*

$$\mathbb{E}_{DP}[\hat{\tau}_\lambda|\boldsymbol{z}] = \hat{\tau}_{\text{No-DP}} \quad \text{and} \quad \mathbb{E}_{DP,\boldsymbol{z}}[\hat{\tau}_\lambda] = \tau$$

Having an unbiased estimator for causal inference is an important but not entirely surprising result. In fact, many differentially private mechanisms can recover true labels in expectation; (Kancharla & Kang, 2021) also propose an unbiased differentially private estimator in the setting of binary potential outcomes $y_i \in \{0,1\}$. Instead, the main difficulty is to minimize the variance gap with non-differentially-private estimators. To state the variance of $\hat{\tau}$ under the UNIFORM-PRIOR-DP mechanism, we consider the following notation: $\bar{y} := 1/|\mathcal{Y}| \sum_{y\in\mathcal{Y}} y$ and $\overline{y^2} := 1/|\mathcal{Y}| \sum_{y\in\mathcal{Y}} y^2$ over all possible outcomes. For $a \in \{0,1\}$, we also define $\overline{y_c(a)} := 1/n_c \sum_{i\in c} y_i(a)$ and $\overline{y_c^2(a)} := 1/n_c \sum_{i\in c} y_i^2(a)$ over the units of cluster $c$.

**Theorem A.5.** *For any $\tilde{\varepsilon} > 0$, the UNIFORM-PRIOR-DP mechanism is $(\tilde{\varepsilon}, \delta)$-label DP when we set $\delta = \max(0, 1 - \lambda + \frac{\lambda}{K}(1 - e^{\tilde{\varepsilon}}))$. In particular, it is $\varepsilon$-label DP with $\varepsilon = \log\left(1 + \frac{(1-\lambda)K}{\lambda}\right)$. Furthermore, the variance of estimator $\hat{\tau}_\lambda$ in (1) under the UNIFORM-PRIOR-DP mechanism and the treatment assignment $\boldsymbol{z}$ is given by*

$$\mathrm{Var}_{DP,\boldsymbol{z}}[\hat{\tau}_\lambda] = \mathrm{Var}_{\boldsymbol{z}}[\hat{\tau}_{\text{No-DP}}] + \sum_{c\in\mathcal{C}} \frac{n_c^2}{n^2}\left(\frac{1}{n_{0,c}} + \frac{1}{n_{1,c}}\right)\frac{\lambda\overline{y^2} - \lambda^2\bar{y}^2}{(1-\lambda)^2}$$

$$+ \sum_{c\in\mathcal{C}} \frac{n_c^2}{n^2}\left[\frac{\lambda}{1-\lambda}\left(\frac{\overline{y_c^2(0)}}{n_{0,c}} + \frac{\overline{y_c^2(1)}}{n_{1,c}}\right) - \frac{2\lambda\bar{y}}{1-\lambda}\left(\frac{\overline{y_c(0)}}{n_{0,c}} + \frac{\overline{y_c(1)}}{n_{1,c}}\right)\right], \quad (7)$$

As the sampling probability grows small $\lambda \to 0$, we recover the non-private variance formula $\mathrm{Var}_{DP,\boldsymbol{z}}(\hat{\tau}) \to \mathrm{Var}_{\boldsymbol{z}}[\hat{\tau}_{\text{No-DP}}]$, but the $\varepsilon$-DP guarantee goes to infinity. Since the UNIFORM-PRIOR-DP mechanism itself does not depend on the clusters, the privacy guarantee does not depend on the clustering properties of the data, if any. The dependence on the clustering in Equation (7) is only due to the definition of the stratified estimator. When a good clustering is not available, the above estimator can be simplified to its unstratified version $\hat{\tau}^u$ by considering that all units belong to the same cluster:

$$\hat{\tau}^u = \frac{1}{1-\lambda}\sum_{i=1}^n \left(\frac{\tilde{y}_i z_i}{n_1} - \frac{\tilde{y}_i(1-z_i)}{n_0}\right). \quad (8)$$

The following variance result is a direct corollary of Theorem A.5.

**Corollary A.6.** *Under the* UNIFORM-PRIOR-DP *mechanism, the variance of the unstratified estimator $\hat{\tau}^u$ defined in Eq. 8 is given by*

$$\text{Var}_{DP,\boldsymbol{z}}[\hat{\tau}^u] = \text{Var}_{\boldsymbol{z}}[\hat{\tau}^u_{\text{No-DP}}] + \frac{n}{n_1 n_0}\frac{\lambda\overline{\mathbf{y}^2} - \lambda^2\bar{\mathbf{y}}^2}{(1-\lambda)^2} + \frac{\lambda}{1-\lambda}\left(\frac{\overline{y^2(0)}}{n_0} + \frac{\overline{y^2(1)}}{n_1}\right) - \frac{2\lambda\bar{\mathbf{y}}}{1-\lambda}\left(\frac{\overline{y(0)}}{n_0} + \frac{\overline{y(1)}}{n_1}\right)$$

*where* $\text{Var}_{\boldsymbol{z}}[\hat{\tau}^u_{\text{No-DP}}]$ *denotes the variance of its non-private equivalent* $\hat{\tau}^u_{\text{No-DP}}$. *Its differential privacy guarantees are the same as those in Theorem A.5.*

We refer the reader to Appendix A.2 for the well-known closed form formula of $\text{Var}_{\boldsymbol{z}}[\hat{\tau}^u_{\text{No-DP}}]$. The special case of the unstratified estimator $\hat{\tau}^u$ in (8) for binary outcomes $\mathcal{Y} = \{0, 1\}$ was previously proposed by (Kancharla & Kang, 2021), in which case the variance of the estimator can be further simplified:

$$\text{Var}_{DP,\boldsymbol{z}}[\hat{\tau}^u] = \text{Var}_{\boldsymbol{z}}[\hat{\tau}^u_{\text{No-DP}}] + \frac{n}{n_0 n_1}\frac{\frac{\lambda}{2}(1-\frac{\lambda}{2})}{(1-\lambda)^2} . \tag{9}$$

The first two aggregation-based mechanisms in Section A.4 assumed that a trusted data curator (e.g. a technology company, in the motivating example in Section 2) has access to the true outcomes and computes a differentially private estimate or empirical distribution of these responses. In contrast, the UNIFORM-PRIOR-DP mechanism can be implemented without such a curator: each user can privatize their response before sharing it with the experimenter. In other words, the UNIFORM-PRIOR-DP mechanism provides a local DP guarantee, defined by (Kasiviswanathan et al., 2011), which is stronger than a DP guarantee. That said, in our motivating example, assuming the existence of a trusted curator—the technology company—is more natural than putting the burden of privatizing responses on each individual user.

### A.6. Experiment 4. (Bias and Gaussianity)

We first verify that our CLUSTER-DP estimator $\hat{\tau_Q}$, given in Theorem 3.2, is unbiased and admits an asymptotically Gaussian distribution by plotting the histogram and the qq-plot of $\hat{\tau} - \tau$ in Figures 3 and 4.

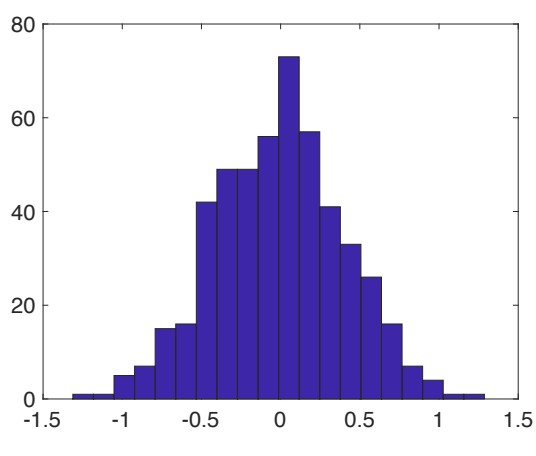

Figure 3: histogram of $\hat{\tau} - \tau$

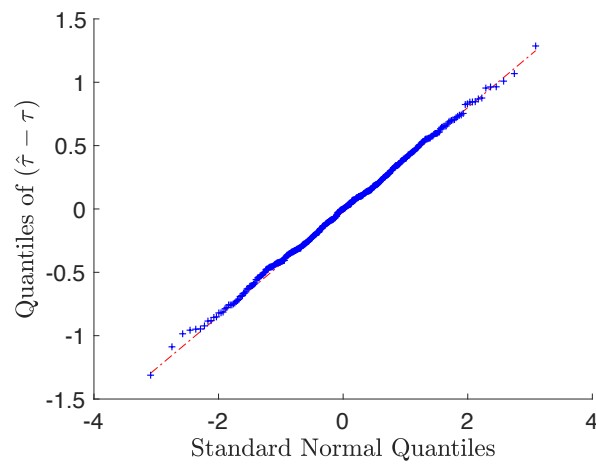

Figure 4: qq-plot of $\hat{\tau} - \tau$

### A.7. Experiment 5. (Validation of theoretical bound)

In Theorem 3.5, we bounded the excess variance of the private estimator given in Theorem 3.2 compared to the non-private estimator (5). The bound had two additive terms. The first one depends on the cluster structure of data, namely the cluster homogeneity quantities $\phi_0, \phi_1$, and the second term did not depend on the clusters, capturing instead an increase in the variance due to the randomness of the CLUSTER-DP mechanism. In Figure 5, we compute the gap $\text{Var}_{DP,\boldsymbol{z}}[\hat{\tau}] - \text{Var}_{\boldsymbol{z}}[\hat{\tau}_{\text{No-DP}}]$ empirically, by averaging over 500 different realizations of the randomness in the DP mechanism and the treatment assignments in the same setting as the previous experiment. We plot this gap as we vary $\beta$, along with

a shaded region whose upper boundary corresponds to the upper bound given in Theorem 3.5 and its lower boundary corresponds to only the first term in that bound. We observe that the variance gap remains in the shaded area which validates the theoretical upper bound given by Theorem 3.5, and shows that the derived bound is tight, up to the second term.

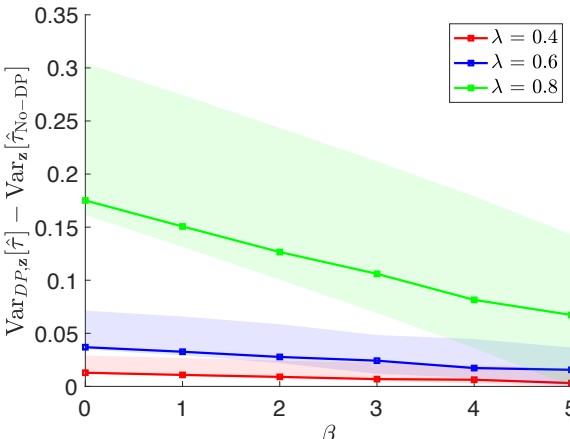

Figure 5: The variance gap between the private estimator $\hat{\tau}_Q$, given given in Theorem 3.2, and the non-private estimator $\hat{\tau}_{\text{No-DP}}$ in the setting of Experiment 5. The upper boundary of the shaded area corresponds to the upper bound derived in Theorem 3.5, and it lower boundary corresponds to the the first term in that bound. As we see the gap remains between the two boundaries.

### A.8. Experiment 6. (Comparisons with aggregation-based baselines)

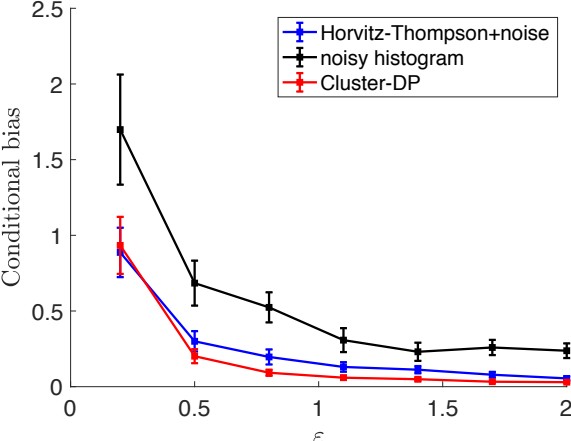

Figure 6: Bias of the CLUSTER-DP, noisy Horvitz-Thompson and noisy histogram estimators under one shot communication between the central unit and the advertisers in the setting of Experiment 6.

We next compare the privacy-variance trade-off of the estimator based on the CLUSTER-DP mechanism with the other baselines discussed in Section 2 and Appendix A.4, namely the noisy Horvitz-Thompson estimator and the noisy histogram estimator. The goal of this experiment is to show that in the case of one-shot communication between the central unit and the advertisers, the CLUSTER-DP estimator achieves lower finite-sample conditional bias than the other two baselines. To demonstrate this point, we fix the noise and randomization in each DP mechanisms for the super-population and compute the bias of each estimator with respect to random draws from the super-population and of the treatment assignments. Specifically we compute the expectation of the treatment effect estimator over 500 sub-populations, each consisting of 500, 1000, 2000 units from each cluster, uniformly at random with a balanced number of treated and controlled units in each cluster. The bias is then computed as the difference between the expectation of the estimator and the true treatment effect. As we see in

Figure 6, CLUSTER-DP estimator achieves a lower conditional bias compared to the other two baselines, as we vary the privacy loss $\varepsilon$. The error bars are obtained by considering 50 different realizations of the noise/randomization in the DP mechanisms.

### A.9. Experiment 7. (Additional qq-plot for YouTube data experiment)

Figure 7 shows the qqplot of $\hat{\tau} - \tau$ with $\hat{\tau}$ being the CLUSTER-DP mechanism, using 500 realizations of the randomness in the outcomes and the DP mechanism, for the YouTube dataset described at the end of Section 4. As the plot demonstrates $\hat{\tau}$ is an unbiased and Gaussian estimator.

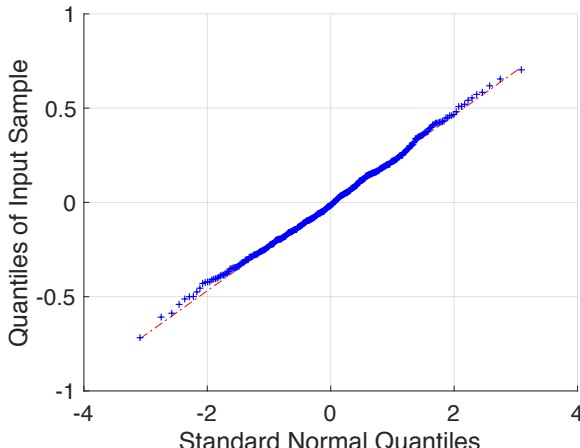

Figure 7: qqplot of $\hat{\tau} - \tau$, with $\hat{\tau}$ the CLUSTER-DP estimator using 500 realizations of randomness in the outcomes and the DP mechanism.

### A.10. Fuller versions of Figures from Experiments 1-3

We include in Figure 8 on the next page fuller versions of the Figures from Experiments 1-3, included in Section 4.

### A.11. Proof of Proposition A.2 and A.3

We start by proving Proposition A.2. Recall the noisy Horvitz-Thompson estimator $\hat{\tau}$ given by (4):

$$\hat{\tau} := \sum_{c \in \mathcal{C}} \frac{n_c}{n} \left\{ \sum_{i \in c} \left( \frac{y_i z_i}{n_{1,c}} - \frac{y_i (1 - z_i)}{n_{0,c}} \right) + w_c \right\} , w_c \sim \text{Laplace}(\eta_c) .$$

To show its privacy guarantee, we apply (Dwork et al. (2014), Theorem 3.6). Consider the sensitivity $\Delta_c$ of the inner function $1/n_{1,c} y_i z_i - 1/n_{0,c} y_i (1 - z_i)$, defined as the maximum change in its value when changing only one label in the data set. Since the assignments are not private, we keep them intact in computing the sensitivity. Therefore, changing only on label will change the inner function by at most $\Delta_c = \min\{n_{0,c}, n_{1,c}\}^{-1} \times \max_{y \in \mathcal{Y}} |y|$. By using (Dwork et al. (2014), Theorem 3.6), adding Laplace noise with parameter $\Delta_c/\varepsilon$ will make each of the inner terms $\varepsilon$-DP and by the post-processing property (Dwork et al. (2014), Proposition 2.1), $\hat{\tau}$ is also $\varepsilon$-DP.

For the variance, recall the non-differentially-private Horvitz-Thompson estimator $\hat{\tau}_{\text{NO-DP}}$ from (5), by which we can write

$$\hat{\tau} = \hat{\tau}_{\text{NO-DP}} + \sum_{c \in \mathcal{C}} \frac{n_c}{n} w_c .$$

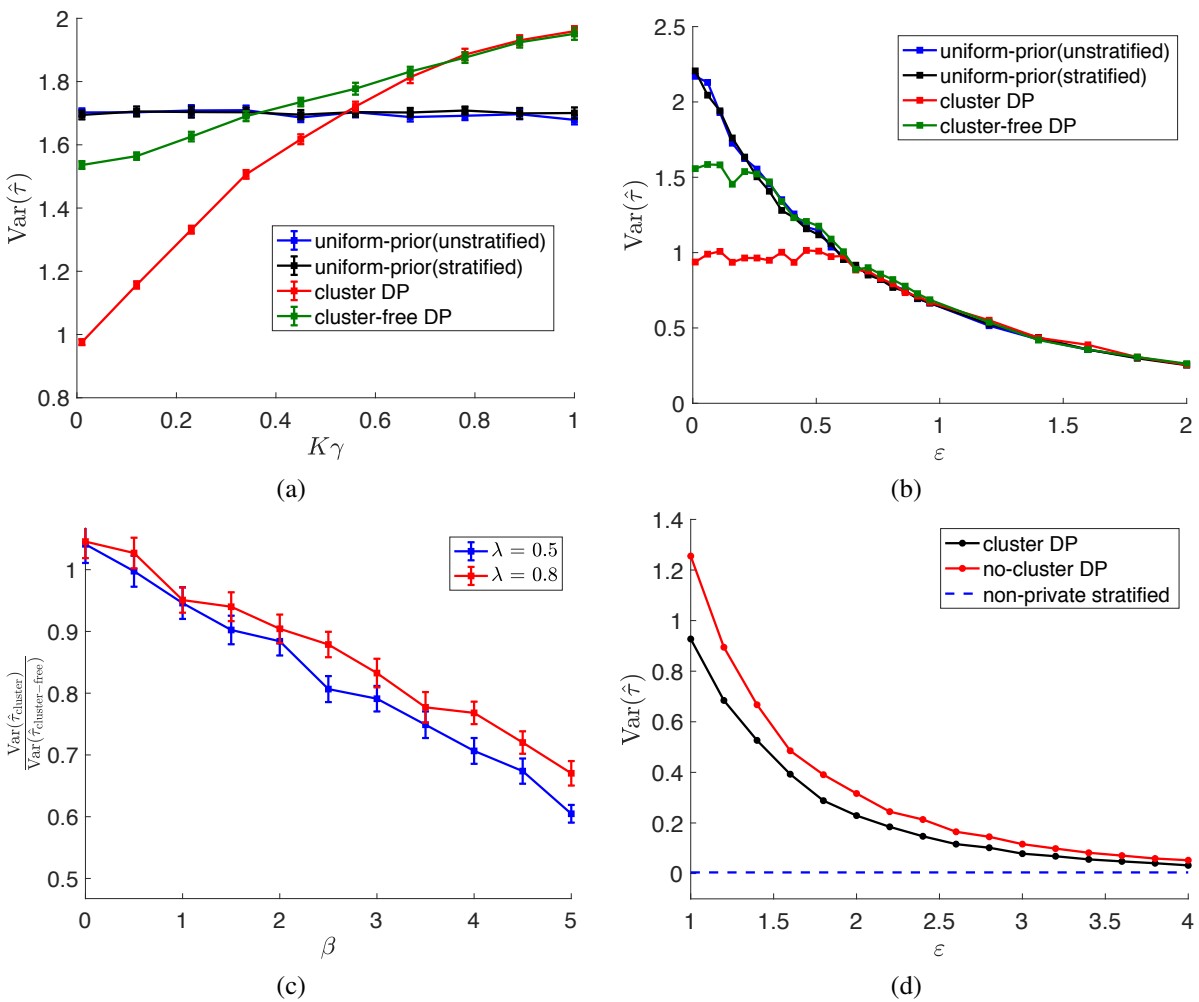

Figure 8: (a) Variances of each mechanism as we vary the truncation level $\gamma \in [0.1/K, 1/K]$ in Experiment 1. The privacy loss is fixed at $\varepsilon = 0.2$ and $\delta = 10^{-4}$. (b) Privacy-variance trade-off of each mechanism under the setting of Experiment 1. We fix the DP failure probability to $\delta = 10^{-4}$, and optimize the choice of $\sigma$ and $\gamma$ in the sets $\sigma \in \{10, 20, \infty\}$ and $\gamma \in \{0.01/K, 0.1/K, 1/K\}$. (c) Ratio of the variance of the estimators under the CLUSTER-DP and CLUSTER FREE-DP mechanisms in Experiment 2. The benefit of CLUSTER-DP mechanism is stronger at larger $\beta$ and smaller value of $\lambda$. (d) Privacy-variance trade-off of the CLUSTER-DP and CLUSTER FREE-DP stratified estimators for the YouTube dataset in Experiment 3. The dotted line represents the variance of the non-private stratified estimator.

Since $w_c$ are drawn independently from each other and also independent from the assignments $z_i$, we have

$$\text{Var}_{DP,\boldsymbol{z}}[\hat{\tau}] = \text{Var}_{DP,\boldsymbol{z}}[\hat{\tau}_{\text{No-DP}}] + \sum_{c\in\mathcal{C}} \left(\frac{n_c}{n}\right)^2 \text{Var}[w_c]$$

$$= \text{Var}_{DP,\boldsymbol{z}}[\hat{\tau}_{\text{No-DP}}] + 2\sum_{c\in\mathcal{C}} \left(\frac{n_c}{n}\frac{\Delta_c}{\varepsilon}\right)^2 ,$$

where the last step holds because $w_c \sim \text{Laplace}(\eta_c)$ with $\eta_c = {\Delta_c}/{\varepsilon}$.

We next proceed with proving Proposition A.3. Its privacy guarantee follows easily from the fact that $\hat{p}_a(y|c)$ has sensitivity $1/n_{a,c}$ (histogram queries) and therefore adding independent draws from $\text{Laplace}((n_{a,c}\varepsilon)^{-1})$ to the frequency of each value will make the histogram $\varepsilon$-DP. To prove the claim on its variance, we note that the non-private Horvitz-Thompson estimator can be written as

$$\hat{\tau}_{\text{No-DP}} = \sum_{c\in\mathcal{C}} \frac{n_c}{n} \left(\sum_{y\in\mathcal{Y}} y\hat{p}_1(y|c) - \sum_{y\in\mathcal{Y}} y\hat{p}_0(y|c)\right) .$$

Therefore, we can write the noisy Histogram estimator (6) as

$$\hat{\tau} = \sum_{c\in\mathcal{C}} \frac{n_c}{n} \sum_{y\in\mathcal{Y}} y(\hat{p}_1(y|c) + w_{1,c,y} - p_0(y|c) - w_{0,c,y})$$

$$= \hat{\tau}_{\text{No-DP}} + \sum_{c\in\mathcal{C}} \frac{n_c}{n} \sum_{y\in\mathcal{Y}} y(w_{1,c,y} - w_{0,c,y}) .$$

Since $w_{a,c,y}$ are independent from each other and $w_{a,c,y} \sim \text{Laplace}(\eta_{a,c})$, we get

$$\text{Var}_{DP,\boldsymbol{z}}[\hat{\tau}] = \text{Var}_{DP,\boldsymbol{z}}[\hat{\tau}_{\text{No-DP}}] + \sum_{c\in\mathcal{C}}\sum_{y\in\mathcal{Y}} \left(\frac{n_c}{n}y\right)^2 (\text{Var}[w_{1,c,y}] + \text{Var}[w_{0,c,y}])$$

$$= \text{Var}_{DP,\boldsymbol{z}}[\hat{\tau}_{\text{No-DP}}] + \left(\sum_{y\in\mathcal{Y}} y^2\right) \sum_{c\in\mathcal{C}} \left(\frac{n_c}{n}\right)^2 (2\eta_{1,c}^2 + 2\eta_{0,c}^2)$$

$$= \text{Var}_{DP,\boldsymbol{z}}[\hat{\tau}_{\text{No-DP}}] + \frac{2}{\varepsilon^2} \left(\sum_{y\in\mathcal{Y}} y^2\right) \sum_{c\in\mathcal{C}} \left(\frac{n_c}{n}\right)^2 \left(\frac{1}{n_{1,c}^2} + \frac{1}{n_{0,c}^2}\right) .$$

This completes the proof of Proposition A.3.

### A.12. Proof of Theorem A.5

Because the UNIFORM-PRIOR DP mechanism is a special case of the CLUSTER-DP mechanism, we follow the proof of Theorem 3.5, which is detailed below and which the reader might prefer reading first. In this special case, we can obtain an exact form for the variance gap. Hence, we continue from (26), which in the case that there is no Laplace noise added, reads as

$$\text{Var}_{DP}(u_{0,c}|\boldsymbol{z},\mathcal{P}) = n_{0,c}\mathsf{y}^\mathsf{T} Q^{-1}\text{diag}(Q\hat{\boldsymbol{p}})Q^{-\mathsf{T}}\mathsf{y} - \sum_{i\in\mathcal{O}_{0,c}} y_i^2(0) . \tag{10}$$

Note that we are using the lightened notation $\hat{\boldsymbol{p}}$ to indicate the empirical distribution of outcomes of the controlled units in cluster $c$.

In the mechanism described by Algorithm 2, $\tilde{\boldsymbol{q}}$ is data-dependent and so correlated to $\hat{\boldsymbol{p}}$. In that case, we analyzed the first term via the decomposition $\text{diag}(Q\hat{\boldsymbol{p}}) = \text{diag}(Q\tilde{\boldsymbol{q}}) + \text{diag}(Q(\hat{\boldsymbol{p}} - \tilde{\boldsymbol{q}}))$ and bounding $\|\tilde{\boldsymbol{q}} - \hat{\boldsymbol{p}}\|_1$. In the current case that $\tilde{\boldsymbol{q}}$ is the uniform distribution, this approach is not tight as $\|\tilde{\boldsymbol{q}} - \hat{\boldsymbol{p}}\|_1$ would be large. However, since $\tilde{\boldsymbol{q}}$ (and therefore $Q$) is

data-independent we can directly analyze the first term as follows:

$$\mathsf{y}^\mathsf{T} Q^{-1}\mathrm{diag}(Q\hat{\boldsymbol{p}})Q^{-\mathsf{T}}\mathsf{y}$$

$$= \frac{1}{(1-\lambda)^2}\mathsf{y}^\mathsf{T}\left(I - \frac{\lambda}{K}\mathbf{1}\mathbf{1}^\mathsf{T}\right)\mathrm{diag}\left((1-\lambda)\hat{\boldsymbol{p}} + \frac{\lambda}{K}\mathbf{1}\right)\left(I - \frac{\lambda}{K}\mathbf{1}\mathbf{1}^\mathsf{T}\right)\mathsf{y}$$

$$= \mathsf{y}^\mathsf{T}\left\{\frac{1}{(1-\lambda)}\mathrm{diag}(\hat{\boldsymbol{p}}) + \frac{\lambda}{K(1-\lambda)^2}\mathrm{diag}(\mathbf{1}) - \frac{2\lambda}{K(1-\lambda)}\mathbf{1}\hat{\boldsymbol{p}}^\mathsf{T} - \frac{\lambda^2}{K^2(1-\lambda)^2}\mathbf{1}\mathbf{1}^\mathsf{T}\right\}\mathsf{y}$$

$$= \frac{\overline{\boldsymbol{y}_{0,c}^2}}{(1-\lambda)} + \frac{\lambda\overline{y^2}}{(1-\lambda)^2} - \frac{2\lambda\bar{y}}{(1-\lambda)}\overline{\boldsymbol{y}_{0,c}} - \frac{\lambda^2\bar{y}^2}{(1-\lambda)^2}$$

$$= \frac{\overline{\boldsymbol{y}_{0,c}^2}}{(1-\lambda)} + \frac{\lambda\overline{y^2} - \lambda^2\bar{y}^2}{(1-\lambda)^2} - \frac{2\lambda\bar{y}}{(1-\lambda)}\overline{\boldsymbol{y}_{0,c}}, \tag{11}$$

where we use the shorthand

$$\overline{\boldsymbol{y}_{0,c}^2} = \frac{1}{n_{0,c}}\sum_{i\in\mathcal{O}_{0,c}} y_i^2(0), \quad \overline{\boldsymbol{y}_{0,c}} = \frac{1}{n_{0,c}}\sum_{i\in\mathcal{O}_{0,c}} y_i(0).$$

Using (11) in (10), we arrive at

$$\mathrm{Var}_{DP}(u_{0,c}|\boldsymbol{z},\mathcal{P}) = n_{0,c}\left[\frac{\overline{\boldsymbol{y}_{0,c}^2}}{(1-\lambda)} + \frac{\lambda\overline{y^2} - \lambda^2\bar{y}^2}{(1-\lambda)^2} - \frac{2\lambda\bar{y}}{(1-\lambda)}\overline{\boldsymbol{y}_{0,c}}\right] - n_{0,c}\overline{\boldsymbol{y}_{0,c}^2}$$

$$= n_{0,c}\left[\frac{\lambda}{1-\lambda}\overline{\boldsymbol{y}_{0,c}^2} + \frac{\lambda\overline{y^2} - \lambda^2\bar{y}^2}{(1-\lambda)^2} - \frac{2\lambda\bar{y}}{(1-\lambda)}\overline{\boldsymbol{y}_{0,c}}\right].$$

Invoking (22), the above characterization yields the following:

$$\mathrm{Var}_{DP}(\hat{\tau}|\boldsymbol{z},\mathcal{P}) = \sum_{c\in\mathcal{C}}\frac{n_c^2}{n^2}\left[\frac{\lambda}{1-\lambda}\left(\frac{\overline{\boldsymbol{y}_{0,c}^2}}{n_{0,c}} + \frac{\overline{\boldsymbol{y}_{1,c}^2}}{n_{1,c}}\right) - \frac{2\lambda\bar{y}}{(1-\lambda)}\left(\frac{\overline{\boldsymbol{y}_{0,c}}}{n_{0,c}} + \frac{\overline{\boldsymbol{y}_{1,c}}}{n_{1,c}}\right)\right]$$

$$+ \sum_{c\in\mathcal{C}}\frac{n_c^2}{n^2}\left(\frac{1}{n_{0,c}} + \frac{1}{n_{1,c}}\right)\frac{\lambda\overline{y^2} - \lambda^2\bar{y}^2}{(1-\lambda)^2}. \tag{12}$$

We next compute $\mathbb{E}_{\boldsymbol{z}}[\mathrm{Var}_{DP}(\hat{\tau}|\boldsymbol{z},\mathcal{P})]$. Since we are fixing $n_{a,c}$ for each cluster, we have $\mathbb{P}(z_i = a) = \frac{n_{a,c}}{n_c}$ for $i\in\mathcal{O}_c$ and $a\in\{0,1\}$. We therefore have

$$\mathbb{E}_{\boldsymbol{z}}[\overline{\boldsymbol{y}_{a,c}}] = \mathbb{E}_{\boldsymbol{z}}\left[\frac{1}{n_{a,c}}\sum_{i\in\mathcal{O}_c}\mathbb{I}(z_i = a)y_i(a)\right] = \frac{1}{n_c}\sum_{i\in\mathcal{O}_c} y_i(a) = \overline{\vec{y}_c(a)}.$$

Likewise we have $\mathbb{E}_{\boldsymbol{z}}[\overline{\boldsymbol{y}_{a,c}^2}] = \overline{\vec{y}_c^2(a)}$. Using this identities in (12), we obtain

$$\mathbb{E}_{\boldsymbol{z}}[\mathrm{Var}_{DP}(\hat{\tau}|\boldsymbol{z},\mathcal{P})] = \sum_{c\in\mathcal{C}}\frac{n_c^2}{n^2}\left[\frac{\lambda}{1-\lambda}\left(\frac{\overline{\vec{y}_c^2(0)}}{n_{0,c}} + \frac{\overline{\vec{y}_c^2(1)}}{n_{1,c}}\right) - \frac{2\lambda\bar{y}}{(1-\lambda)}\left(\frac{\overline{\vec{y}_c(0)}}{n_{0,c}} + \frac{\overline{\vec{y}_{c(1)}}}{n_{1,c}}\right)\right]$$

$$+ \sum_{c\in\mathcal{C}}\frac{n_c^2}{n^2}\left(\frac{1}{n_{0,c}} + \frac{1}{n_{1,c}}\right)\frac{\lambda\overline{y^2} - \lambda^2\bar{y}^2}{(1-\lambda)^2}.$$

We next recall (21):

$$\mathrm{Var}_{\boldsymbol{z}}[\mathbb{E}_{DP}(\hat{\tau}|\boldsymbol{z},\mathcal{P})] = \sum_{c\in\mathcal{C}}\frac{n_c^2}{n^2}\left(\frac{S^2(\vec{y}_c(1))}{n_{1,c}} + \frac{S^2(\vec{y}_c(0))}{n_{0,c}} - \frac{S^2(\vec{\tau}_c)}{n_c}\right),$$

which is the variance of the typical estimator with no-differential-privacy and so was written as $\mathrm{Var}_{\boldsymbol{z}}[\hat{\tau}_{\text{No-DP}}]$. Finally, from the law of total variance, we have:

$$
\begin{aligned}
\mathrm{Var}(\hat{\tau}|n_0, n_1, \mathcal{P}) &= \mathbb{E}_{\boldsymbol{z}}[\mathrm{Var}_{DP}(\hat{\tau}|\boldsymbol{z}, n_0, n_1, \mathcal{P})] + \mathrm{Var}_{\boldsymbol{z}}[\mathbb{E}_{DP}(\hat{\tau}|\boldsymbol{z}, n_0, n_1, \mathcal{P})] \\
&= \mathbb{E}_{\boldsymbol{z}}[\mathrm{Var}_{DP}(\hat{\tau}|\boldsymbol{z}, \mathcal{P})] + \mathrm{Var}_{\boldsymbol{z}}[\mathbb{E}_{DP}(\hat{\tau}|\boldsymbol{z}, \mathcal{P})] \\
&= \mathrm{Var}_{\boldsymbol{z}}[\hat{\tau}_{\text{No-DP}}] + \sum_{c \in \mathcal{C}} \frac{n_c^2}{n^2} \left( \frac{1}{n_{0,c}} + \frac{1}{n_{1,c}} \right) \frac{\lambda \overline{y^2} - \lambda^2 \bar{y}^2}{(1-\lambda)^2} \\
&\quad + \sum_{c \in \mathcal{C}} \frac{n_c^2}{n^2} \left[ \frac{\lambda}{1-\lambda} \left( \frac{\overline{\boldsymbol{y}_c^2(0)}}{n_{0,c}} + \frac{\overline{\boldsymbol{y}_c^2(1)}}{n_{1,c}} \right) - \frac{2\lambda \bar{y}}{(1-\lambda)} \left( \frac{\overline{\boldsymbol{y}_c(0)}}{n_{0,c}} + \frac{\overline{\boldsymbol{y}_c(1)}}{n_{1,c}} \right) \right].
\end{aligned}
$$

## A.13. Proof of Theorem 3.1

The CLUSTER-DP mechanism randomizes the labels using the empirical probability of units with the same treatment status (treated or controlled) within the same cluster, so we can focus on the controlled units within one cluster, and drop the index $a, c$ from our notation, unless needed for clarification. With slight abuse of notation, suppose that there are $n$ controlled units in the cluster and denote by $M$ the mechanism described in Algorithm 2.

We can think of $M$ as composition of two mechanisms $M_1$ and $M_2$ with $M(D) = M_2(D, M_1(D))$, where $M_1(D)$ represents the mechanism that returns the noisy cluster label distribution $\tilde{q}$, and $M_2(D, \tilde{q})$ represents the mechanism which uses $\tilde{q}$ to re-sample the labels and use them to form the average treatment effect estimator $\hat{\tau}$. By composition theorem for $(\varepsilon, \delta)$-DP (see e.g. (Dwork et al. (2014), Theorem B.1), if $M_1$ is $(\varepsilon_1, \delta_1)$-DP and $M_2$ is $(\varepsilon_2, \delta_2)$-DP, then $M$ is $(\varepsilon_1 + \varepsilon_2, \delta_1 + \delta_2)$-DP.

After adding noise terms $w_{y,c}$ to empirical distributions $\hat{p}_{0,D}(y|c)$ and $\hat{p}_{1,D}(y|c)$, the dataset $D$ is not accessed anymore. Furthermore, the empirical distributions have sensitivity $1/n_c$ and the Laplace noise used in $M_1$ is of scale $\sigma/n_c$, which imply that $M_1$ is $(1/\sigma, 0)$-DP (see e.g. (Dwork et al. (2014), Theorem 3.6) for an argument).

For mechanism $M_2$, note that it is a randomization per label mechanism (using perturbed distribution $\tilde{q}$), followed by post-processing (computing average treatment effect estimator). We next show that $M_2$ is $(\tilde{\varepsilon}, \delta)$-DP. Note that for all $y \in \mathcal{Y}$, we have

$$
\mathbb{P}(\tilde{y}_i = y|y_i = y) = 1 - \lambda + \lambda \tilde{q}(y), \quad \mathbb{P}(\tilde{y}_i = y|y_i \neq y) = \lambda \tilde{q}(y).
$$

Since $\mathbb{P}(\tilde{y}_i = y|y_i \neq y)$ is independent of $y_i$ and $\mathbb{P}(\tilde{y}_i = y|y_i \neq y) < \mathbb{P}(\tilde{y}_i = y|y_i = y)$, the only condition we need to verify is the following:

$$
\mathbb{P}(\tilde{y}_i = y|y_i = y) \leq e^{\tilde{\varepsilon}} \mathbb{P}(\tilde{y}_i = y|y_i \neq y) + \delta.
$$

By substituting for the events probabilities, the above condition becomes

$$
1 - \lambda + \lambda \tilde{q}(y) \leq \lambda \tilde{q}(y) e^{\tilde{\varepsilon}} + \delta.
$$

By rearranging the terms, it can be rewritten as

$$
1 - \lambda + \lambda \tilde{q}(y)(1 - e^{\tilde{\varepsilon}}) \leq \delta.
$$

Now recall that $\delta := \max(0, 1 - \lambda + \lambda \gamma (1 - e^{\tilde{\varepsilon}}))$. Hence, it is sufficient to show that

$$
1 - \lambda + \lambda \tilde{q}(y)(1 - e^{\tilde{\varepsilon}}) \leq 1 - \lambda + \lambda \gamma (1 - e^{\tilde{\varepsilon}}),
$$

which by rearranging the terms reads as

$$
0 \leq \lambda(\gamma - \tilde{q}(y))(1 - e^{\tilde{\varepsilon}}),
$$

which holds since $\tilde{\varepsilon} > 0$ and $\gamma \leq \tilde{q}(y)$. To summarize, by applying composition theorem for $(\varepsilon, \delta)$-DP, we obtain that $M$ is $(\varepsilon', \delta)$-label DP, with $\varepsilon' = 1/\sigma + \tilde{\varepsilon}$.

We next show that $M$ is $(\varepsilon'', \delta)$-label DP, with $\varepsilon'' = 2/\gamma + \sigma$, which along with the previous result gives the claim of Theorem 3.1. Let $Y_{1:n}$ be the random vector denoting the labels of the units. We need to show that for any two neighboring data sets $D = (y_{1:n}, x_{1:n})$ and $D' = (y'_{1:n}, x_{1:n})$ (where $y_{1:n}$ and $y'_{1:n}$ differ only in one entry) we have

$$
\mathbb{P}(M(y_{1:n}) \in O) \leq e^{\varepsilon''} \mathbb{P}(M(y'_{1:n}) \in O) + \delta, \tag{13}
$$

for any set $O \in \mathcal{Y}^n$. Proof of this part requires more effort. Let $W_{1:K}$ be the random vector representing the noise values added to the set of possible labels in the data set. We then have

$$
\begin{aligned}
\mathbb{P}(M(y_{1:n}) \in O) &= \sum_{w_{1:K}} \mathbb{P}(M(y_{1:n}) \in O)|Y_{1:n} = y_{1:n}, W_{1:K} = w_{1:K}) \, \mathbb{P}(W_{1:K} = w_{1:K}) \, , \\
\mathbb{P}(M(y'_{1:n}) \in O) &= \sum_{w_{1:K}} \mathbb{P}(M(y'_{1:n}) \in O)|Y_{1:n} = y'_{1:n}, W_{1:K} = w_{1:K}) \, \mathbb{P}(W_{1:K} = w_{1:K}) \, .
\end{aligned}
\tag{14}
$$

It suffices to show that for any value of $w_{1:K}$, we have

$$
\begin{aligned}
\mathbb{P}(M(y_{1:n}) \in O)|Y_{1:n} = y_{1:n}, W_{1:K} = w_{1:K}) & \tag{15} \\
\leq e^{\varepsilon''} \mathbb{P}(M(y'_{1:n}) \in O)|Y_{1:n} = y'_{1:n}, W_{1:K} = w_{1:K}) + \delta \, .
\end{aligned}
$$

By multiplying both sides of the above equation with $\mathbb{P}(W_{1:K} = w_{1:K})$, and summing over $w_{1:K}$, and using that $\sum_{w_{1:K}} \mathbb{P}(W_{1:K} = w_{1:K}) = 1$, we get the desired bound in (13).

Let $\tilde{q}$ and $\tilde{q}'$ be the empirical distributions of the DP mechanism, as defined in Algorithm 2. we continue by establishing a lemma on $\|\tilde{q} - \tilde{q}'\|_\infty$, proven in the next section.

**Lemma A.7.** *For all* $y \in \mathcal{Y}$, $|\tilde{q}(y) - \tilde{q}'(y)| \leq \frac{2}{n}$ .

Define the shorthand $R := M(y_{1:n})$ and $R' := M(y'_{1:n})$. In order to prove (15), it suffices to show that for all $o_{1:n} \in \mathcal{Y}^n$, we have

$$
\mathbb{P}(R = o_{1:n}|Y_{1:n} = y_{1:n}, W_{1:K} = w_{1:K}) \leq e^{\varepsilon''} \mathbb{P}(R' = o_{1:n}|Y_{1:n} = y'_{1:n}, W_{1:K} = w_{1:K}) + \delta \, .
\tag{16}
$$

By the definition of the mechanism $M$ we have

$$
\begin{aligned}
\mathbb{P}(R = o_{1:n}|Y_{1:n} = y_{1:n}, W_{1:K} = w_{1:K}) &= \mathbb{P}(R = o_{1:n}|Y_{1:n} = y_{1:n}, \tilde{q}(\cdot)) \\
&= \prod_{i=1}^{n} \mathbb{P}(R_i = o_i|Y_i = y_i, \tilde{q}(\cdot)) \\
\mathbb{P}(R' = o_{1:n}|Y_{1:n} = y'_{1:n}, W_{1:K} = w_{1:K}) &= \mathbb{P}(R = o_{1:n}|Y_{1:n} = y'_{1:n}, \tilde{q}'(\cdot)) \\
&= \prod_{i=1}^{n} \mathbb{P}(R'_i = o_i|Y'_i = y'_i, \tilde{q}'(\cdot))
\end{aligned}
$$

For ease in presentation, we adopt the shorthand

$$
A_i := \mathbb{P}(R_i = o_i|Y_i = y_i, \tilde{q}(\cdot)), \quad B_i := \mathbb{P}(R'_i = o_i|Y'_i = y'_i, \tilde{q}'(\cdot)) \, ,
$$

for $i = 1, \ldots, n$. Our next lemma bounds the event probability $A_i$ in terms of the event probability $B_i$. Proof of Lemma A.8 is deferred to Section A.13.

**Lemma A.8.** *Let* $\tilde{\varepsilon} > 0$ *and define* $\delta := (1 - \lambda + \lambda\gamma(1 - e^{\tilde{\varepsilon}}))_+ < 1$. *Without loss of generality suppose that the neighboring label sets* $y_{1:n}$ *and* $y'_{1:n}$ *differs in the first coordinate. We then have*

$$
A_1 \leq e^{\tilde{\varepsilon}} \left(1 + \frac{2}{\gamma n}\right) B_1 + \delta,
\tag{17}
$$

$$
A_i \leq B_i \left(1 + \frac{2}{\gamma n}\right), \quad \text{for } i = 2, \ldots, n.
\tag{18}
$$

We are now ready to prove inequality (16). Using Lemma A.8, we write

$$\mathbb{P}(R = o_{1:n}|Y_{1:n} = y_{1:n}, W_{1:K} = w_{1:K}) = \prod_{i=1}^{n} A_i = A_1 \min\left\{1, \prod_{i=2}^{n} A_i\right\}$$

$$\leq \left[e^{\tilde{\varepsilon}}\left(1 + \frac{2}{\gamma n}\right)B_1 + \delta\right]\min\left\{1, \left(1 + \frac{2}{\gamma n}\right)^{n-1}\prod_{i=2}^{n} B_i\right\}$$

$$\leq e^{\tilde{\varepsilon}}\left(1 + \frac{2}{\gamma n}\right)^{n}\prod_{i=1}^{n} B_i + \delta$$

$$\leq e^{\tilde{\varepsilon}+2/\gamma}\prod_{i=1}^{n} B_i + \delta$$

$$= e^{\varepsilon''}\mathbb{P}(R' = o_{1:n}|Y_{1:n} = y'_{1:n}, W_{1:K} = w_{1:K}) + \delta.$$

where the second equality holds since $A_i \leq 1$, for all $i$.

**Proof of Lemma A.7**

Recall the notation of Theorem 3.1. We consider two neighboring datasets $D = (y_{1:n}, x_{1:n})$ and $D' = (y'_{1:n}, x_{1:n})$, where $y_{1:n}$ and $y'_{1:n}$ differ only in one entry. Define the function $f_\gamma$ as follows:

$$f_\gamma(x) = \max\{\gamma, \min\{1, x\}\} = \begin{cases} \gamma, & x \leq \gamma \\ x, & \gamma \leq x \leq 1 \\ 1, & x > 1 \end{cases}$$

We consider $q(y) := f_\gamma(\hat{p}(y) + w_y)$ and $q'(y) := f_\gamma(\hat{p}'(y) + w_y)$, where $\hat{p}$ and $\hat{p}'$ respectively denote the empirical distribution of $y_{1:n}$ and $y'_{1:n}$ and $w_y$ indicates the component of $w_{1:K}$ corresponding to label $y$. We wish to bound the difference between distributions $\tilde{q}(y)$ and $\tilde{q}'(y)$, defined in Algorithm 2, and recalled below:

$$\tilde{q}(y) = q(y) + \frac{\zeta_y}{\sum_{y'} \zeta_{y'}}\Delta, \quad \tilde{q}'(y) = q'(y) + \frac{\zeta'_y}{\sum_{y'} \zeta'_{y'}}\Delta',$$

where $\Delta = 1 - \sum_y q(y)$ and $\Delta' = 1 - \sum_y q'(y)$. To achieve this, we will need a bound on $|q(y) - q'(y)|$ and on a bound on $|\Delta - \Delta'|$.

- Since $f_\gamma$ is 1-Lipschitz, we have for any $y \in \mathcal{Y}$

$$|q(y) - q'(y)| = |f_\gamma(\hat{p}(y) + w_y) - f_\gamma(\hat{p}'(y) + w_y)| \leq |\hat{p}(y) - \hat{p}'(y)| \leq \frac{1}{n},$$

where the last inequality holds because the datasets $D$ and $D'$ differ in only one label.

- We now show that $|\Delta - \Delta'| \leq 1/n$. Without loss of generality, we can assume that the neighboring label sets $y_{1:n}$ and $y'_{1:n}$ differ in the first coordinate, with $y_1 = \ell, y'_1 = \ell'$ for $\ell, \ell' \in \mathcal{Y}$, such that

$$\hat{p}(\ell) = \hat{p}'(\ell) + \frac{1}{n}, \quad \hat{p}(\ell') = \hat{p}'(\ell') - \frac{1}{n}.$$

It follows that

$$\Delta' - \Delta = \sum_y q(y) - \sum_y q'(y)$$

$$= f_\gamma(\hat{p}(\ell) + w_\ell) + f_\gamma(\hat{p}(\ell') + w_{\ell'}) - f_\gamma(\hat{p}'(\ell) + w_\ell) - f_\gamma(\hat{p}'(\ell') + w_{\ell'})$$

$$= f_\gamma\left(\hat{p}'(\ell) + \frac{1}{n} + w_\ell\right) - f_\gamma(\hat{p}'(\ell) + w_\ell) + f_\gamma\left(\hat{p}'(\ell') - \frac{1}{n} + w_{\ell'}\right) - f_\gamma(\hat{p}'(\ell') + w_{\ell'})$$

$$\leq f_\gamma\left(\hat{p}'(\ell) + \frac{1}{n} + w_\ell\right) - f_\gamma(\hat{p}'(\ell) + w_\ell) \leq \frac{1}{n},$$

where the second to last inequality holds since $f_\gamma$ is a non-decreasing function, and the last step follows from 1-Lipschitzness of $f_\gamma$. Likewise, we can show $\Delta - \Delta' \leq 1/n$ in order to obtain $|\Delta - \Delta'| \leq 1/n$.

With this, we next bound the difference between distributions $\tilde{q}(y)$ and $\tilde{q}'(y)$, defined above. Consider three different cases:

- $\Delta > 0, \Delta' < 0$. We have

$$|\tilde{q}(y) - \tilde{q}'(y)| \leq |q(y) - q'(y)| + \left| \frac{\zeta_y}{\sum_{y'} \zeta_{y'}} \Delta - \frac{\zeta'_y}{\sum_{y'} \zeta'_{y'}} \Delta' \right|$$

$$\leq |q(y) - q'(y)| + |\Delta - \Delta'| \leq \frac{2}{n}.$$

The case of $\Delta < 0, \Delta' > 0$ can be handled similarly.

- $\Delta, \Delta' < 0$. We have

$$\tilde{q}(y) = q(y) + (q(y) - \gamma) \frac{\Delta}{\sum_{y'} (q(y') - \gamma)}$$

$$= q(y) + (q(y) - \gamma) \frac{\Delta}{1 - \Delta - K\gamma}$$

$$= \gamma + (q(y) - \gamma) + (q(y) - \gamma) \frac{\Delta}{1 - \Delta - K\gamma}$$

$$= \gamma + (q(y) - \gamma) \frac{1 - K\gamma}{1 - \Delta - K\gamma}.$$

Therefore,

$$|\tilde{q}(y) - \tilde{q}'(y)| \leq (q(y) - \gamma) \left| \frac{1 - K\gamma}{1 - \Delta - K\gamma} - \frac{1 - K\gamma}{1 - \Delta' - K\gamma} \right| + |q'(y) - q(y)| \frac{1 - K\gamma}{1 - \Delta' - K\gamma}$$

$$= (q(y) - \gamma) \frac{(1 - K\gamma)|\Delta - \Delta'|}{(1 - \Delta - K\gamma)(1 - \Delta' - K\gamma)} + \frac{1}{n} \frac{1 - K\gamma}{1 - \Delta' - K\gamma}$$

$$= \frac{1 - K\gamma}{1 - \Delta' - K\gamma} \left[ \frac{(q(y) - \gamma)|\Delta - \Delta'|}{(1 - \Delta - K\gamma)} + \frac{1}{n} \right]$$

$$\overset{(a)}{\leq} \frac{1}{n} \frac{1 - K\gamma}{1 - \Delta' - K\gamma} \left( \frac{q(y) - \gamma}{1 - \Delta - K\gamma} + 1 \right)$$

$$\overset{(b)}{\leq} \frac{2}{n} \frac{1 - K\gamma}{1 - \Delta' - K\gamma} \leq \frac{2}{n}.$$

$(a)$ holds since $|\Delta - \Delta'| \leq 1/n$. $(b)$ follows from the fact that, since $q(y) \geq \gamma$ for all $y$,

$$q(y) + (K - 1)\gamma \leq q(y) + \sum_{y' \neq y} q(y') = 1 - \Delta,$$

such that $q(y) - \gamma \leq 1 - \Delta - K\gamma$.

- $\Delta, \Delta' > 0$. We have

$$\tilde{q}(y) = q(y) + (1 - q(y)) \frac{\Delta}{\sum_{y'} (1 - q(y'))}$$

$$= 1 + (1 - q(y)) \left( \frac{\Delta}{\Delta + K - 1} - 1 \right)$$

$$= 1 + (1 - q(y)) \frac{1 - K}{\Delta + K - 1}.$$

Therefore,

$$|\tilde{q}(y) - \tilde{q}'(y)| \leq (1 - q(y)) \left| \frac{1-K}{\Delta + K - 1} - \frac{1-K}{\Delta' + K - 1} \right| + |q(y) - q'(y)| \frac{K-1}{\Delta' + K - 1}$$

$$\leq (1 - q(y)) \left[ \frac{|\Delta' - \Delta|}{\Delta + K - 1} + \frac{1}{n} \right] \frac{K-1}{\Delta' + K - 1}$$

$$\leq (1 - q(y)) \frac{1}{n(K-1)} + \frac{1}{n} \leq \frac{1}{n} \frac{K}{K-1} \leq \frac{2}{n} \, .$$

Combining the above three cases together, we obtain our stated lemma.

### Proof of Lemma A.8

We start by proving (17). Consider three different cases:

- $o_1 \neq y_1, y_1'$: In this case, we have $A_1 = \lambda \tilde{q}(o_1)$, $B_1 = \lambda \tilde{q}'(o_1)$. Therefore, we can write

$$A_1 \leq \lambda \tilde{q}'(o_1) + \lambda \|\tilde{q} - \tilde{q}'\|_\infty$$

$$\leq \lambda \tilde{q}'(o_1) + \frac{2\lambda}{n}$$

$$\leq \lambda \tilde{q}'(o_1) \left( 1 + \frac{2}{n\gamma} \right)$$

$$= B_1 \left( 1 + \frac{2}{n\gamma} \right)$$

$$\leq e^{\tilde{\varepsilon}} B_1 \left( 1 + \frac{2}{n\gamma} \right) + \delta,$$

where the second step follows from Lemma A.7, third step holds since $\gamma \leq \tilde{q}'(o_1)$, and the last step holds since $\tilde{\varepsilon}, \delta > 0$. So the claim (17) is proved in this case.

- $o_1 = y_1$: In this case, $A_1 = 1 - \lambda + \lambda \tilde{q}(o_1)$, $B_1 = \lambda \tilde{q}'(o_1)$. We then have

$$A_1 \leq 1 - \lambda + \lambda \tilde{q}'(o_1) + \lambda \|\tilde{q} - \tilde{q}'\|_\infty$$

$$\leq 1 - \lambda + \lambda \tilde{q}'(o_1) + \lambda \frac{2}{n\gamma} \tilde{q}'(o_1) e^{\tilde{\varepsilon}}, \qquad (19)$$

where we used Lemma A.7 along with the facts that $\tilde{q}'(o_1) \geq \gamma$ and $\tilde{\varepsilon} > 0$.

We next recall the definition $\delta := (1 - \lambda + \lambda\gamma(1 - e^{\tilde{\varepsilon}}))_+ < 1$. By a simple rearrangement of the terms and using that $\tilde{q}'(o_1) \geq \gamma$ and $\tilde{\varepsilon} > 0$, we can verify the following,

$$1 - \lambda + \lambda \tilde{q}'(o_1) \leq e^{\tilde{\varepsilon}} \lambda \tilde{q}'(o_1) + \delta \, . \qquad (20)$$

Therefore, by combining equations (19) and (20), we get

$$A_1 \leq e^{\tilde{\varepsilon}} \lambda \tilde{q}'(o_1) + \delta + \lambda \frac{2}{n\gamma} \tilde{q}'(o_1) e^{\tilde{\varepsilon}} = e^{\tilde{\varepsilon}} = \left( 1 + \frac{2}{n\gamma} \right) B_1 + \delta \, ,$$

which completes the proof of claim (17) in this case.

- $o_1 = y_1'$: In this case, $A_1 = \lambda \tilde{q}(o_1)$, $B_1 = 1 - \lambda + \lambda \tilde{q}'(o_1)$. The proof of claim (17) in this case follows readily from case 1, because $A_1$ is the same as in there, while $B_1$ is larger.

This concludes the proof of Claim 17. We next prove Claim 18. Note that for $i = 2, \ldots, n$, we have $y_i = y_i'$. Consider the following two cases:

- $o_i = y_i = y_i'$: In this case we have $\frac{A_i}{B_i} = \frac{1 - \lambda + \lambda \tilde{q}(o_i)}{1 - \lambda + \lambda \tilde{q}'(o_i)} \, .$

- $o_i \neq y_i$: Since $y_i = y'_i$, we also have $o_i \neq y'_i$. In this case, $\frac{A_i}{B_i} = \frac{\lambda \tilde{q}(o_i)}{\lambda \tilde{q}'(o_i)}$.

By symmetry, we can assume $\tilde{q}(o_i) \leq \tilde{q}'(o_i)$, without loss of generality, and therefore, the maximum value of the ratio $A_1/B1$ is achieved in the second case, for which we have $\frac{A_i}{B_i} = \frac{\tilde{q}(o_i)}{\tilde{q}'(o_i)} \leq 1 + \frac{\|\tilde{q} - \tilde{q}'\|_\infty}{\tilde{q}'(o_i)} \leq 1 + \frac{2}{n\gamma}$. This completes the proof of (18).

## A.14. Proof of Theorem 3.2

We would like to express the expectation $\mathbb{E}(\hat{\tau}|n_0, n_1)$. Recall that there are three sources of randomness:

- the differential privacy mechanism $DP$: determines the Laplace noise $\boldsymbol{w}$ and the $\lambda$ probability of reporting the true outcome.

- the randomized assignment $\boldsymbol{z}$: determines which units get assigned to treatment and which units get assigned to control.

- the super-population $\mathcal{P}$: determines the potential outcomes as well as the cluster assignments.

For a given unit $i$ with $(y_i(0), y_i(1), c_i) \sim \mathcal{P}$ and $z_i = a$,

$$\mathbb{E}_{DP}\left[\sum_{y' \in \mathcal{Y}} Q_{c_i, z_i}^{-1}[y', \tilde{y}_i]y' z_i \bigg| \boldsymbol{z}, \mathcal{P}\right] = \mathbb{E}_{DP}\left[\sum_{y' \in \mathcal{Y}} \sum_{y \in \mathcal{Y}} \mathbb{I}(\tilde{y}_i = y)Q_{c_i, z_i}^{-1}[y', y]y' z_i \bigg| \boldsymbol{z}, \mathcal{P}\right]$$

$$\overset{(a)}{=} \sum_{y' \in \mathcal{Y}} \mathbb{E}_{DP}\left[\sum_{y \in \mathcal{Y}} \mathbb{I}(\tilde{y}_i = y)Q_{c_i, z_i}^{-1}[y', y] \bigg| \boldsymbol{z}, \mathcal{P}\right] y' z_i$$

$$\overset{(b)}{=} \sum_{y' \in \mathcal{Y}} \mathbb{E}_{\boldsymbol{w}}\left[\sum_{y \in \mathcal{Y}} \mathbb{E}_\lambda\left[\mathbb{I}(\tilde{y}_i = y)\right] Q_{c_i, z_i}^{-1}[y', y] \bigg| \boldsymbol{z}, \mathcal{P}\right] y' z_i$$

$$\overset{(c)}{=} \sum_{y' \in \mathcal{Y}} \mathbb{E}_{\boldsymbol{w}}\left[\sum_{y \in \mathcal{Y}} Q_{c_i, z_i}[y, y_i]Q_{c_i, z_i}^{-1}[y', y] \bigg| \boldsymbol{z}, \mathcal{P}\right] y' z_i$$

$$\overset{(d)}{=} \sum_{y' \in \mathcal{Y}} \mathbb{E}_{\boldsymbol{w}}\left[\mathbb{I}(y_i = y')|\boldsymbol{z}, \mathcal{P}\right] y' z_i$$

$$\overset{(e)}{=} \sum_{y' \in \mathcal{Y}} \mathbb{I}(y_i = y')y' z_i = y_i(1)z_i.$$

$(a)$ holds since assignments $z_i$ is independent from $\{y_i(0), y_i(1), c_i\}$; $(b)$ holds from the law of iterated expectation and the fact that there are two sources of randomness in the differential privacy mechanism: $(\lambda, \boldsymbol{w})$ with $Q_{c,a}$ independent of the Bernoulli $\lambda$; $(c)$ follows from the definition of $Q_{c,a}$: $Q_{c,a}[y', y] = (1 - \lambda)\mathbb{I}(y' = y) + \lambda \tilde{q}_a(y'|c)$; and $(d)$ follows from the fact that $I = Q_{c_i}^{-1}Q_{c_i}$ therefore, for any $a, b \in [K]$,

$$\sum_y Q_{c_i}^{-1}[a, y]Q_{c_i}[y, b] = I_{a,b} = \mathbb{I}(a = b).$$

Finally, $(e)$ follows from the fact that $\boldsymbol{w}$ is independent from $\{y_i(0), y_i(1), c_i\}$. Similarly,

$$\mathbb{E}_{DP}\left[\sum_{y' \in \mathcal{Y}} Q_{c_i, z_i}^{-1}[y', \tilde{y}_i]y'(1 - z_i) \bigg| \boldsymbol{z}, \mathcal{P}\right] = y_i(0)(1 - z_i)$$

As a result, with $n_{0,c}$ (resp. $n_{1,c}$) the total number of controlled (resp. treated) units in cluster $c$ and $n_c := n_{0,c} + n_{1,c}$,

$$\mathbb{E}_{DP}[\hat{\tau}|\boldsymbol{z}, \mathcal{P}] = \sum_{c \in \mathcal{C}} \frac{n_c}{n}\left(\sum_{i=1}^n y_i(1)\frac{z_i}{n_{1,c}} - \sum_{i=1}^n y_i(0)\frac{1 - z_i}{n_{0,c}}\right)$$

We recover the standard form of the difference-in-means estimator. From the law of iterated expectations, we have

$$\mathbb{E}_{DP,\boldsymbol{z}}[\hat{\tau}] = \mathbb{E}_{\boldsymbol{z}}\left[\mathbb{E}_{DP}[\hat{\tau}|\boldsymbol{z}]|n_0, n_1, \mathcal{P}\right] = \tau.$$

### A.15. Proof of Theorem 3.5

We would like to express the variance $\mathrm{Var}_{DP,\boldsymbol{z}}(\hat{\tau})$. We begin by expressing the variance with respect to the first two, considering the third fixed. From the law of total variance, we have:

$$\mathrm{Var}_{DP,\boldsymbol{z}}(\hat{\tau}) = \mathbb{E}_{\boldsymbol{z}}[\mathrm{Var}_{DP}(\hat{\tau}|\boldsymbol{z}, n_0, n_1, \mathcal{P})] + \mathrm{Var}_{\boldsymbol{z}}[\mathbb{E}_{DP}(\hat{\tau}|\boldsymbol{z}, n_0, n_1, \mathcal{P})]$$
$$= \mathbb{E}_{\boldsymbol{z}}[\mathrm{Var}_{DP}(\hat{\tau}|\boldsymbol{z}, \mathcal{P})] + \mathrm{Var}_{\boldsymbol{z}}[\mathbb{E}_{DP}(\hat{\tau}|\boldsymbol{z}, \mathcal{P})]$$

We bound the term $\mathrm{Var}_{DP}(\hat{\tau}|\boldsymbol{z}, \mathcal{P})$ in a separate proposition

**Proposition A.9.** *For the average treatment effect estimator $\hat{\tau}_Q$ given given in Theorem 3.2, we have*

$$\mathrm{Var}_{DP}(\hat{\tau}|\boldsymbol{z}, \mathcal{P}) \leq \sum_{a \in \{0,1\}} \sum_{c \in \mathcal{C}} \frac{n_c^2}{n^2} \left[\left(\frac{1}{(1-\lambda)^2} - 1\right) \frac{S^2(\vec{y}_{a,c})}{n_{a,c}} + \frac{A(n_{a,c})}{n_{a,c}}\right],$$

*where $\vec{y}_{a,c} := \{y_i(a) : i \in \mathcal{O}_{a,c}\}$, and*

$$A(x) = 2KB^2 \left(\left(\frac{3}{(1-\lambda)^2} + 2\right) + \frac{(\lambda\sqrt{K}+1)^2}{(1-\lambda)^2} \|\mathbf{y}\|^2 (1 - \lambda(K-1)\gamma)\right) \left[\gamma + \frac{\sigma}{x}\left(e^{-\gamma x/\sigma} - e^{-x/\sigma}\right)\right]$$

We now take its expectation with respect to $\boldsymbol{z}$. We assume that, for each cluster c, there is a fixed number of units $(n_{1,c})$ assigned to treatment and a fixed number of units $(n_{0,c})$ assigned to control, regardless of the cluster assignment. We compute the expectation with $\mathbb{E}_{\boldsymbol{z}}\left[S^2(\vec{y}_{a,c})\right]$,

$$(n_{a,c} - 1)\mathbb{E}_{\boldsymbol{z}}\left[S^2(\vec{y}_{a,c})\right]$$

$$= \mathbb{E}_{\boldsymbol{z}}\left[\sum_{i \in \mathcal{O}_{a,c}} \left(y_i(a) - \overline{\{y_i(a)\}_{i \in \mathcal{O}_{a,c}}}\right)^2\right]$$

$$= \mathbb{E}_{\boldsymbol{z}}\left[\sum_{i \in \mathcal{O}_{a,c}} y_i^2(a) - n_{a,c}\left(\frac{1}{n_{a,c}} \sum_{i \in \mathcal{O}_{a,c}} y_i(a)\right)^2\right]$$

$$= \sum_{i \in \mathcal{O}_c} \mathbb{P}(z_i = a) y_i^2(a) - n_{a,c}\mathbb{E}_{\boldsymbol{z}}\left[\left(\frac{1}{n_{a,c}} \sum_{i \in \mathcal{O}_c} \mathbb{I}(z_i = a)y_i(a)\right)^2\right]$$

$$= \sum_{i \in \mathcal{O}_c} \frac{n_{a,c}}{n_c} y_i^2(a) - \frac{1}{n_{a,c}} \sum_{i \in \mathcal{O}_c} \sum_{j \in \mathcal{O}_c} \mathbb{P}(z_i = a, z_j = a) y_i(a)y_j(a)$$

$$= \sum_{i \in \mathcal{O}_c} \frac{n_{a,c}}{n_c} y_i^2(a) - \frac{1}{n_{a,c}} \sum_{i \in \mathcal{O}_c} \mathbb{P}(z_i = a) y_i^2(a) - \frac{1}{n_{a,c}} \sum_{j \neq i \in \mathcal{O}_c} \mathbb{P}(z_i = a, z_j = a) y_i(a)y_j(a)$$

$$= \sum_{i \in \mathcal{O}_c} \frac{n_{a,c}}{n_c} y_i^2(a) - \frac{1}{n_c} \sum_{i \in \mathcal{O}_c} y_i^2(a) - \frac{1}{n_{a,c}} \sum_{j \neq i \in \mathcal{O}_c} \frac{n_{a,c}(n_{a,c} - 1)}{n_c(n_c - 1)} y_i(a)y_j(a).$$

Adding and subtracting $\frac{n_{a,c}-1}{n_c(n_c-1)} \sum_{i \in \mathcal{O}_c} y_i^2(a)$, we get:

$$(n_{a,c} - 1)\mathbb{E}_{\boldsymbol{z}} \left[ S^2(\vec{y}_{a,c}) \right]$$

$$= \sum_{i \in \mathcal{O}_c} \frac{n_{a,c}}{n_c} y_i^2(a) - \frac{1}{n_c} \sum_{i \in \mathcal{O}_c} y_i^2(a) + \frac{n_{a,c}-1}{n_c(n_c-1)} \sum_{i \in \mathcal{O}_c} y_i^2(a) - \frac{n_{a,c}-1}{n_c(n_c-1)} \left( \sum_{i \in \mathcal{O}_c} y_i(a) \right)^2$$

$$= \left( \frac{n_{a,c}}{n_c} - \frac{1}{n_c} + \frac{n_{a,c}-1}{n_c(n_c-1)} \right) \sum_{i \in \mathcal{O}_c} y_i^2(a) - (n_{a,c} - 1) \frac{n_c}{n_c - 1} \left( \frac{1}{n_c} \sum_{i \in \mathcal{O}_c} y_i(a) \right)^2$$

$$= \frac{n_{a,c}-1}{n_c - 1} \sum_{i \in \mathcal{O}_c} y_i^2(a) - (n_{a,c} - 1) \frac{n_c}{n_c - 1} \left( \frac{1}{n_c} \sum_{i \in \mathcal{O}_c} y_i(a) \right)^2$$

$$= (n_{a,c} - 1) S^2(\vec{y}_c(a)).$$

For the second term, we again make the assumption that the number of treated units is fixed at the cluster level. For the second term, from the proof of Theorem 3.2, we have:

$$\mathbb{E}_{DP}[\hat{\tau}|\boldsymbol{z}, \mathcal{P}] = \sum_{c \in \mathcal{C}} \frac{n_c}{n} \left( \sum_{i=1}^{n} y_i(1) \frac{z_i}{n_{1,c}} - \sum_{i=1}^{n} y_i(0) \frac{1 - z_i}{n_{0,c}} \right)$$

As a result, the second term is given by the usual formula for the variance of the stratified estimator:

$$\text{Var}_{\boldsymbol{z}}[\mathbb{E}_{DP}(\hat{\tau}|\boldsymbol{z}, \mathcal{P})] = \sum_{c \in \mathcal{C}} \frac{n_c^2}{n^2} \left( \frac{S^2(\vec{y}_c(1))}{n_{1,c}} + \frac{S^2(\vec{y}_c(0))}{n_{0,c}} - \frac{S^2(\vec{\tau}_c)}{n_c} \right) \tag{21}$$

where, for any vector $\vec{u}$ of length $n$, $S^2(u) = \frac{1}{n-1} \sum_{i=1}^{n} (u_i - \bar{u})^2$ and $\bar{u} = \frac{1}{n} \sum_{i=1}^{n} u_i$. Recall that $\vec{\tau}_c = \{y_i(1) - y_i(0)\}_{i:c_i=c} = \vec{y}_c(1) - \vec{y}_c(0)$. Since this is the variance of the typical estimator with no-differential-privacy, we write this term:

$$\text{Var}_{\boldsymbol{z}}[\mathbb{E}_{DP}(\hat{\tau}|\boldsymbol{z}, \mathcal{P})] = \text{Var}_{\boldsymbol{z}}[\hat{\tau}_{\text{No-DP}}]$$

As a result, we obtain

$$\text{Var}_{DP,\boldsymbol{z}}(\hat{\tau}|n_{0,c}, n_{1,c}, \mathcal{P})$$

$$\leq \text{Var}_{\boldsymbol{z}}[\hat{\tau}_{\text{No-DP}}] + \sum_{a \in \{0,1\}} \sum_{c \in \mathcal{C}} \frac{n_c^2}{n^2} \left[ \left( \frac{1}{(1-\lambda)^2} - 1 \right) \frac{S^2(\vec{y}_c(a))}{n_{a,c}} + \frac{A(n_{a,c})}{n_{a,c}} \right]$$

$$\leq \text{Var}_{\boldsymbol{z}}[\hat{\tau}_{\text{No-DP}}] + \left( \frac{1}{(1-\lambda)^2} - 1 \right) \sum_{a \in \{0,1\}} \sum_{c \in \mathcal{C}} \frac{n_c^2}{n^2} \frac{S^2(\vec{y}_c(a))}{n_{a,c}} + \sum_{a \in \{0,1\}} \sum_{c \in \mathcal{C}} \frac{n_c^2}{n^2} \frac{A(n_{a,c})}{n_{a,c}}$$

which we can rewrite as:

$$\text{Var}_{DP,\boldsymbol{z}}(\hat{\tau}|n_{0,c}, n_{1,c}, \mathcal{P}) \leq \text{Var}_{\boldsymbol{z}}[\hat{\tau}_{\text{No-DP}}] + \left( \frac{1}{(1-\lambda)^2} - 1 \right) \sum_{a \in \{0,1\}} \phi_a + \sum_{a \in \{0,1\}} \sum_{c \in \mathcal{C}} \frac{n_c^2}{n^2} \frac{A(n_{a,c})}{n_{a,c}}$$

where we have defined

$$\phi_a := \sum_{c \in \mathcal{C}} \frac{n_c^2}{n^2} \frac{S^2(\vec{y}_c(a))}{n_{a,c}} \geq 0$$

### Proof of Proposition A.9

We seek to compute $\text{Var}_{DP}(\hat{\tau}|\boldsymbol{z}, \mathcal{P})$. We can rewrite $\hat{\tau}$ as

$$\hat{\tau} = \sum_{c \in C} \frac{n_c}{n} \sum_{a \in \{0,1\}} \frac{u_{a,c}}{n_{a,c}},$$

where $\mathcal{O}_{a,c} := \{i \in [n] : c_i = c, z_i = a\}$ and $u_{a,c} := \sum_{i \in \mathcal{O}_{a,c}} \sum_{y' \in \mathcal{Y}} Q_{c_i,a}^{-1}[y', \tilde{y}_i] y'$. Since $(y_i(0), y_i(1))$ are i.i.d across units, and the DP mechanism is applied to each clusters separately, such that the privatized outcomes $\tilde{y}_i$ are independent across clusters, we have that $u_{0,c}$ and $u_{1,c}$ are independent across clusters.

$$\mathrm{Var}_{DP}(\hat{\tau}|\boldsymbol{z}, \mathcal{P}) = \sum_{c \in \mathcal{C}} \frac{n_c^2}{n^2} \sum_{a \in \{0,1\}} \frac{1}{n_{a,c}^2} \mathrm{Var}_{DP}(u_{a,c}|\boldsymbol{z}, \mathcal{P}) . \tag{22}$$

We proceed by calculating $\mathrm{Var}_{DP}(u_{0,c}|\boldsymbol{z}, \mathcal{P})$. The computation for $\mathrm{Var}_{DP}(u_{1,c}|\boldsymbol{z}, \mathcal{P})$ is identical.

• COMPUTING $\mathrm{Var}_{DP}(u_{0,c}|\boldsymbol{z}, \mathcal{P})$

We have
$$\mathrm{Var}_{DP}(u_{0,c}|\boldsymbol{z}, \mathcal{P}) = \mathbb{E}_{DP}[u_{0,c}^2|\boldsymbol{z}, \mathcal{P}] - \mathbb{E}_{DP}[u_{0,c}|\boldsymbol{z}, \mathcal{P}]^2$$

We begin by computing $\mathbb{E}_{DP}[u_{0,c}|\boldsymbol{z}, \mathcal{P}]$. Fixing cluster $c$, we lighten the notation by using the shorthand $Q = Q_{c,0}$, $Q_{a,b} = Q_{c,0}[a, b]$, $Q_{a,b}^{-1} = Q_{c,0}^{-1}[a, b]$, and $Q^{-\mathsf{T}} = (Q^{-1})^{\mathsf{T}}$. Finally, recall that $\mathsf{y} = (y)_{y \in \mathcal{Y}}$ is the set of possible outcomes arranged into a vector with the same ordering as the columns of $Q$, and $e_\ell \in \mathbb{R}^{K \times 1}$ is the vector with 1 at the $\ell$-th position and zero everywhere else. Writing in matrix form, we have

$$u_{0,c} = \sum_{i \in \mathcal{O}_{0,c}} \mathsf{y}^{\mathsf{T}} Q_{\cdot,\tilde{y}_i}^{-1} .$$

Let $\tilde{\boldsymbol{y}}, \boldsymbol{y}, \boldsymbol{z}, \boldsymbol{w}$ be the vectors of variables $\tilde{y}_i, y_i, z_i, (w)_{y,c}$ respectively. We then have

$$\mathbb{E}_{DP}[u_{0,c}|\boldsymbol{z}, \mathcal{P}] = \sum_{i \in \mathcal{O}_{0,c}} \mathbb{E}_{DP}\left[\mathsf{y}^{\mathsf{T}} Q_{\cdot,\tilde{y}_i}^{-1} \Big| \boldsymbol{z}, \mathcal{P}\right]$$

$$= \sum_{i \in \mathcal{O}_{0,c}} \mathbb{E}_{DP}\left[\mathsf{y}^{\mathsf{T}} \sum_{\ell \in \mathcal{Y}} Q_{\cdot,\ell}^{-1} \mathbb{I}(l = \tilde{y}_i) \Big| \boldsymbol{z}, \mathcal{P}\right]$$

$$= \sum_{i \in \mathcal{O}_{0,c}} \mathsf{y}^{\mathsf{T}} \sum_{\ell \in \mathcal{Y}} \mathbb{E}_{DP}\left[Q_{\cdot,\ell}^{-1} \mathbb{I}(l = \tilde{y}_i) \Big| \boldsymbol{z}, \mathcal{P}\right]$$

Following similar steps to the proof of Theorem 3.2, we have

$$\mathbb{E}_{DP}\left[Q_{\cdot,\ell}^{-1} \mathbb{I}(l = \tilde{y}_i) \Big| \boldsymbol{z}, \mathcal{P}\right] = \mathbb{E}_{\boldsymbol{w}}\left[Q_{\cdot,\ell}^{-1} \mathbb{E}_\lambda\left[\mathbb{I}(l = \tilde{y}_i)|\boldsymbol{w}\right] \Big| \boldsymbol{z}, \mathcal{P}\right] = \mathbb{E}_{\boldsymbol{w}}\left[Q_{\cdot,\ell}^{-1} Q_{l,y_i} \Big| \boldsymbol{z}, \mathcal{P}\right] = I e_i .$$

It follows
$$\mathbb{E}_{DP}[u_{0,c}|\boldsymbol{z}, \mathcal{P}] = \sum_{i \in \mathcal{O}_{0,c}} \mathsf{y}^{\mathsf{T}} I e_i = \sum_{i \in \mathcal{O}_{0,c}} y_i(0) . \tag{23}$$

We next calculate $\mathbb{E}[u_{0,c}^2|\boldsymbol{z}, \mathcal{P}]$.

$$\mathbb{E}_{DP}\left[u_{0,c}^2|\boldsymbol{z}, \mathcal{P}\right] = \sum_{i,j \in \mathcal{O}_{0,c}} \mathbb{E}_{DP}\left[\mathsf{y}^{\mathsf{T}} Q_{\cdot,\tilde{y}_j}^{-1} (Q_{\cdot,\tilde{y}_i}^{-1})^{\mathsf{T}} \mathsf{y} \Big| \boldsymbol{z}, \mathcal{P}\right]$$

$$= \sum_{i,j \in \mathcal{O}_{0,c}} \mathsf{y}^{\mathsf{T}} \mathbb{E}_{DP}\left[\sum_{l,l' \in \mathcal{Y}} Q_{\cdot,l'}^{-1} (Q_{\cdot,l}^{-1})^{\mathsf{T}} \mathbb{I}(l = \tilde{y}_i) \mathbb{I}(l' = \tilde{y}_j) \Big| \boldsymbol{z}, \mathcal{P}\right] \mathsf{y}$$

$$= \sum_{i,j \in \mathcal{O}_{0,c}} \mathsf{y}^{\mathsf{T}} \mathbb{E}_{\boldsymbol{w}}\left[\sum_{l,l' \in \mathcal{Y}} Q_{\cdot,l'}^{-1} (Q_{\cdot,l}^{-1})^{\mathsf{T}} \mathbb{E}_\lambda\left[\mathbb{I}(l = \tilde{y}_i) \mathbb{I}(l' = \tilde{y}_j)|\boldsymbol{w}\right] \Big| \boldsymbol{z}, \mathcal{P}\right] \mathsf{y}$$

$$= \sum_{i \in \mathcal{O}_{0,c}} \mathsf{y}^{\mathsf{T}} \mathbb{E}_{\boldsymbol{w}}\left[\sum_{\ell \in \mathcal{Y}} Q_{\cdot,\ell}^{-1} (Q_{\cdot,\ell}^{-1})^{\mathsf{T}} Q_{\ell,y_i} \Big| \boldsymbol{z}, \mathcal{P}\right] \mathsf{y}$$

$$+ \sum_{i \neq j \in \mathcal{O}_{0,c}} \mathsf{y}^{\mathsf{T}} \mathbb{E}_{\boldsymbol{w}}\left[\sum_{\ell,\ell' \in \mathcal{Y}} Q_{\cdot,\ell'}^{-1} (Q_{\cdot,\ell}^{-1})^{\mathsf{T}} Q_{\ell,y_i} Q_{\ell',y_j} \Big| \boldsymbol{z}, \mathcal{P}\right] \mathsf{y} \tag{24}$$

For the first term, we write

$$\sum_{\ell \in \mathcal{Y}} Q_{.,\ell}^{-1} (Q_{.,\ell}^{-1})^\mathsf{T} Q_{\ell, y_i} = \sum_{\ell \in \mathcal{Y}} Q_{.,\ell}^{-1} (Q_{.,\ell}^{-1})^\mathsf{T} (Q e_{y_i})_\ell = Q^{-1} \mathrm{diag}(Q e_{y_i}) Q^{-\mathsf{T}}, \tag{25}$$

Let $\hat{\boldsymbol{p}} = [\hat{p}_0(y|c)]_{y \in \mathcal{Y}} \in \mathbb{R}^{K \times 1}$ be the empirical distributions of outcomes of the controlled units in cluster $c$, arranged into a $K$-dimensional vector, such that $\hat{p}_\ell = \frac{1}{n_{0,c}} |\{i \in \mathcal{O}_{0,c} : y_i = \ell\}|$. In vector form,

$$\hat{\boldsymbol{p}} = \frac{1}{n_{0,c}} \sum_{i \in \mathcal{O}_{0,c}} e_{y_i}.$$

Taking the expectation of both sides in (25) and summing over $i \in \mathcal{O}_{0,c}$, we get

$$\sum_{i \in \mathcal{O}_{0,c}} \mathbb{E}_{\boldsymbol{w}} \left[ \sum_{\ell \in \mathcal{Y}} Q_{.,\ell}^{-1} (Q_{.,\ell}^{-1})^\mathsf{T} Q_{\ell, y_i} \,\middle|\, \boldsymbol{z}, \mathcal{P} \right] = \mathbb{E}_{\boldsymbol{w}} \left[ Q^{-1} \mathrm{diag} \left( Q \sum_{i \in \mathcal{O}_{0,c}} e_{y_i} \right) Q^{-\mathsf{T}} \,\middle|\, \boldsymbol{z}, \mathcal{P} \right]$$

$$= \mathbb{E}_{\boldsymbol{w}} \left[ Q^{-1} \mathrm{diag}(n_{0,c} Q \hat{\boldsymbol{p}}) Q^{-\mathsf{T}} \,\middle|\, \boldsymbol{z}, \mathcal{P} \right]$$

$$= n_{0,c} \mathbb{E}_{\boldsymbol{w}} \left[ Q^{-1} \mathrm{diag}(Q \hat{\boldsymbol{p}}) Q^{-\mathsf{T}} \,\middle|\, \boldsymbol{z}, \mathcal{P} \right]$$

We next proceed with the second term on the right-hand side of (24). We have

$$\sum_{\ell, \ell' \in \mathcal{Y}} Q_{.,\ell'}^{-1} Q_{.,\ell}^{-\mathsf{T}} Q_{\ell, y_i} Q_{\ell', y_j} = \sum_{\ell, \ell' \in \mathcal{Y}} Q_{.,\ell'}^{-1} Q_{.,\ell}^{-\mathsf{T}} (Q e_{y_j})_{\ell'} (Q e_{y_i})_\ell$$

$$= \sum_{\ell, \ell' \in \mathcal{Y}} Q_{.,\ell'}^{-1} [(Q e_{y_j})_{\ell'} (Q e_{y_i})_\ell] Q_{.,\ell}^{-\mathsf{T}}$$

$$= \sum_{\ell, \ell' \in \mathcal{Y}} Q_{.,\ell'}^{-1} (Q e_{y_j} e_{y_i}^\mathsf{T} Q^\mathsf{T})_{\ell', \ell} Q_{.,\ell}^{-\mathsf{T}}$$

$$= Q^{-1} Q e_{y_j} e_{y_i}^\mathsf{T} Q^\mathsf{T} Q^{-T} = e_{y_j} e_{y_i}^\mathsf{T}.$$

Taking the expectation of both sides of the above equation, we arrive at

$$\mathsf{y}^\mathsf{T} \mathbb{E}_{\boldsymbol{w}} \left[ \sum_{\ell, \ell' \in \mathcal{Y}} Q_{.,\ell'}^{-1} (Q_{.,\ell}^{-1})^\mathsf{T} Q_{\ell, y_i} Q_{\ell', y_j} \,\middle|\, \boldsymbol{z}, \mathcal{P} \right] \mathsf{y} = \mathbb{E}_{\boldsymbol{w}} \left[ \mathsf{y}^\mathsf{T} e_{y_j} e_{y_i}^\mathsf{T} \mathsf{y} \,\middle|\, \boldsymbol{z}, \mathcal{P} \right]$$

$$= y_i(0) y_j(0),$$

where the second equality holds since $i \neq j \in \mathcal{O}_{0,c}$. Putting these pieces together, we obtain

$$\mathbb{E}_{DP}[u_{0,c}^2 | \boldsymbol{z}, \mathcal{P}] = n_{0,c} \mathsf{y}^\mathsf{T} \mathbb{E}_{\boldsymbol{w}} \left[ Q^{-1} \mathrm{diag}(Q \hat{\boldsymbol{p}}) Q^{-\mathsf{T}} \,\middle|\, \boldsymbol{z}, \mathcal{P} \right] \mathsf{y} + \sum_{i \neq j \in \mathcal{O}_{0,c}} y_i(0) y_j(0),$$

which along with (23) gives us

$$\mathrm{Var}_{DP}(u_{0,c} | \boldsymbol{z}, \mathcal{P})$$

$$= \mathbb{E}_{DP}[u_{0,c}^2 | \boldsymbol{z}, \mathcal{P}] - \mathbb{E}_{DP}[u_{0,c} | \boldsymbol{z}, \mathcal{P}]^2$$

$$= n_{0,c} \mathsf{y}^\mathsf{T} \mathbb{E}_{\boldsymbol{w}} \left[ Q^{-1} \mathrm{diag}(Q \hat{\boldsymbol{p}}) Q^{-\mathsf{T}} | \boldsymbol{z}, \mathcal{P} \right] \mathsf{y} + \sum_{i \neq j \in \mathcal{O}_{o,c}} y_i(0) y_j(0) - \left( \sum_{i \in \mathcal{O}_{0,c}} y_i(0) \right)^2$$

$$= n_{0,c} \mathsf{y}^\mathsf{T} \mathbb{E}_{\boldsymbol{w}} \left[ Q^{-1} \mathrm{diag}(Q \hat{\boldsymbol{p}}) Q^{-\mathsf{T}} \,\middle|\, \boldsymbol{z}, \mathcal{P} \right] \mathsf{y} - \sum_{i \in \mathcal{O}_{0,c}} y_i^2(0). \tag{26}$$

• DECOMPOSING $\text{Var}_{DP}(u_{0,c}|\boldsymbol{z}, \mathcal{P})$

We wish to bound $\text{Var}(u_{0,c}|\boldsymbol{z}, \mathcal{P})$. We begin by decomposing it into two distinct terms. Let $\tilde{\boldsymbol{q}} = [\tilde{q}_0(y|c)]_{y \in \mathcal{Y}} \in \mathbb{R}^{K \times 1}$ be the distribution constructed in the DP mechanism after adding noise to the empirical distribution $\hat{\boldsymbol{p}}$, truncation, and normalization. We consider the following decomposition:

$$Q^{-1}\text{diag}(Q\hat{\boldsymbol{p}})Q^{-\mathsf{T}} = Q^{-1}\text{diag}(Q\tilde{\boldsymbol{q}})Q^{-\mathsf{T}} + Q^{-1}\text{diag}(Q(\hat{\boldsymbol{p}} - \tilde{\boldsymbol{q}}))Q^{-\mathsf{T}}.$$

Plugging into (26),

$$\text{Var}_{DP}(u_{0,c}|\boldsymbol{z}, \mathcal{P}) =$$

$$n_{0,c}\left(\underbrace{\mathsf{y}^{\mathsf{T}}\mathbb{E}_{\boldsymbol{w}}\left[Q^{-1}\text{diag}(Q\tilde{\boldsymbol{q}})Q^{-\mathsf{T}}|\boldsymbol{z}, \mathcal{P}\right]\mathsf{y}}_{\text{term I}} + \underbrace{\mathsf{y}^{\mathsf{T}}\mathbb{E}_{\boldsymbol{w}}\left[Q^{-1}\text{diag}(Q(\hat{\boldsymbol{p}} - \tilde{\boldsymbol{q}}))Q^{-\mathsf{T}}|\boldsymbol{z}, \mathcal{P}\right]\mathsf{y}}_{\text{term II}}\right) - \sum_{i \in \mathcal{O}_{0,c}} y_i^2(0).$$

• BOUNDING TERM I

By definition, we can write $Q$ as $Q = (1 - \lambda)I + \lambda\tilde{\boldsymbol{q}}\boldsymbol{1}^{\mathsf{T}}$, with $\boldsymbol{1} \in \mathbb{R}^{K \times 1}$ indicating the all-one vector. Furthermore, $\boldsymbol{1}^{\mathsf{T}}\tilde{\boldsymbol{q}} = 1$ because $\tilde{\boldsymbol{q}}$ is a probability distribution (Esfandiari et al. (2022), Theorem 6). Using the Sherman–Morrison formula, we obtain

$$Q^{-1} = \frac{1}{1 - \lambda}I - \frac{\lambda}{1 - \lambda}\tilde{\boldsymbol{q}}\boldsymbol{1}^{\mathsf{T}}. \tag{27}$$

Plugging for $Q$ and $Q^{-1}$, we have the following chain of identities:

$$Q\tilde{\boldsymbol{q}} = (1 - \lambda)\tilde{\boldsymbol{q}} + \lambda\tilde{\boldsymbol{q}} = \tilde{\boldsymbol{q}},$$

$$Q^{-1}\text{diag}(Q\tilde{\boldsymbol{q}}) = \frac{1}{1 - \lambda}\text{diag}(\tilde{\boldsymbol{q}}) - \frac{\lambda}{1 - \lambda}\tilde{\boldsymbol{q}}\tilde{\boldsymbol{q}}^{\mathsf{T}},$$

$$Q^{-1}\text{diag}(Q\tilde{\boldsymbol{q}})Q^{-T} = \frac{1}{(1 - \lambda)^2}\text{diag}(\tilde{\boldsymbol{q}}) + \frac{\lambda^2 - 2\lambda}{(1 - \lambda)^2}\tilde{\boldsymbol{q}}\tilde{\boldsymbol{q}}^{\mathsf{T}}.$$

Using the last identity, we have

$$\mathsf{y}^{\mathsf{T}}\mathbb{E}_{\boldsymbol{w}}\left[Q^{-1}\text{diag}(Q\tilde{\boldsymbol{q}})Q^{-\mathsf{T}}|\boldsymbol{z}, \mathcal{P}\right]\mathsf{y}$$

$$= \frac{1}{(1 - \lambda)^2}\mathsf{y}^{\mathsf{T}}\mathbb{E}_{\boldsymbol{w}}\left[\text{diag}(\tilde{q})\right]\mathsf{y} + \frac{\lambda^2 - 2\lambda}{(1 - \lambda)^2}\mathsf{y}^{\mathsf{T}}\mathbb{E}_{\boldsymbol{w}}\left[\tilde{\boldsymbol{q}}\tilde{\boldsymbol{q}}^{\mathsf{T}}\right]\mathsf{y}$$

$$= \frac{1}{(1 - \lambda)^2}\sum_{y \in \mathcal{Y}}\mathbb{E}_{\boldsymbol{w}}\left[\tilde{q}_y\right]y^2 + \frac{\lambda^2 - 2\lambda}{(1 - \lambda)^2}\mathbb{E}_{\boldsymbol{w}}\left[\left(\sum_{y \in \mathcal{Y}}y\tilde{q}_y\right)^2\right]$$

$$= \mathbb{E}_{\boldsymbol{w}}\left[\frac{1}{(1 - \lambda)^2}\mathbb{E}_{\tilde{q}}[y_i^2(0)] + \left(1 - \frac{1}{(1 - \lambda)^2}\right)\mathbb{E}_{\tilde{q}}[y_i(0)]^2\Big|\boldsymbol{z}, \mathcal{P}\right]$$

$$= \mathbb{E}_{\boldsymbol{w}}\left[\frac{1}{(1 - \lambda)^2}\left(\mathbb{E}_{\tilde{q}}[y_i^2(0)] - \mathbb{E}_{\tilde{q}}[y_i(0)]^2\right) + E_{\tilde{q}}[y_i(0)]^2\Big|\boldsymbol{z}, \mathcal{P}\right]. \tag{28}$$

which can also be written as:

$$\mathsf{y}^{\mathsf{T}}\mathbb{E}_{\boldsymbol{w}}\left[Q^{-1}\text{diag}(Q\tilde{\boldsymbol{q}})Q^{-\mathsf{T}}|\boldsymbol{z}, \mathcal{P}\right]\mathsf{y} = \frac{1}{(1 - \lambda)^2}\left(\mathbb{E}_{\tilde{q},\boldsymbol{w}}[y_i^2(0)|\boldsymbol{z}, \mathcal{P}] - \mathbb{E}_{\tilde{q},\boldsymbol{w}}[y_i(0)|\boldsymbol{z}, \mathcal{P}]^2\right) + E_{\tilde{q},\boldsymbol{w}}[y_i(0)|\boldsymbol{z}, \mathcal{P}]^2$$

In the following lemma, we relate the expectation of outcomes with respect to $\tilde{\boldsymbol{q}}$ to their expectation with respect to $\hat{\boldsymbol{p}}$.

**Lemma A.10.** *If outcomes are bounded by $B$,*

$$\mathbb{E}_{\tilde{q},\boldsymbol{w}}[y_i^2(0)]|\boldsymbol{z}, \mathcal{P}] \le B^2\mathbb{E}_{\boldsymbol{w}}\|\tilde{\boldsymbol{q}} - \hat{\boldsymbol{p}}\|_1 + \frac{1}{n_{0,c}}\sum_{i \in \mathcal{O}_{0,c}}y_i(0)^2.$$

$$\mathbb{E}_{\tilde{q},\boldsymbol{w}}[y_i(0)|\boldsymbol{z}, \mathcal{P}]^2 \ge \left(\frac{1}{n_{0,c}}\sum_{i \in \mathcal{O}_{0,c}}y_i(0)\right)^2 - 2B^2\mathbb{E}_{\boldsymbol{w}}\|\tilde{\boldsymbol{q}} - \hat{\boldsymbol{p}}\|_1.$$

Using the above lemma, we obtain:

$$\mathsf{y}^\mathsf{T}\mathbb{E}_{\boldsymbol{w}}\left[Q^{-1}\mathrm{diag}(Q\tilde{\boldsymbol{q}})Q^{-\mathsf{T}}\big|\boldsymbol{z},\mathcal{P}\right]\mathsf{y} \leq \frac{1}{(1-\lambda)^2}\left[B^2\mathbb{E}_{\boldsymbol{w}}\|\tilde{\boldsymbol{q}}-\hat{\boldsymbol{p}}\|_1 + \frac{1}{n_{0,c}}\sum_{i\in\mathcal{O}_{0,c}}y_i(0)^2\right]$$

$$-\left(\frac{1}{(1-\lambda)^2}-1\right)\left\{\left(\frac{1}{n_{0,c}}\sum_{i\in\mathcal{O}_{0,c}}y_i(0)\right)^2 - 2B^2\mathbb{E}_{\boldsymbol{w}}\|\tilde{\boldsymbol{q}}-\hat{\boldsymbol{p}}\|_1\right\},$$

which can be simplified to

$$\mathsf{y}^\mathsf{T}\mathbb{E}_{\boldsymbol{w}}\left[Q^{-1}\mathrm{diag}(Q\tilde{\boldsymbol{q}})Q^{-\mathsf{T}}\big|\boldsymbol{z},\mathcal{P}\right]\mathsf{y}$$

$$\leq \frac{1}{(1-\lambda)^2}\left(\frac{1}{n_{0,c}}\sum_{i\in\mathcal{O}_{0,c}}y_i(0)^2 - \left(\frac{1}{n_{0,c}}\sum_{i\in\mathcal{O}_{0,c}}y_i(0)\right)^2\right)$$

$$+\left(\frac{1}{n_{0,c}}\sum_{i\in\mathcal{O}_{0,c}}y_i(0)\right)^2 + B^2\left(\frac{3}{(1-\lambda)^2}+2\right)\mathbb{E}_{\boldsymbol{w}}\|\tilde{\boldsymbol{q}}-\hat{\boldsymbol{p}}\|_1$$

$$\leq \frac{n_{0,c}-1}{n_{0,c}}\frac{S^2(\vec{y}_{0,c})}{(1-\lambda)^2} + \left(\overline{\vec{y}_c(0)}\right)^2 + B^2\left(\frac{3}{(1-\lambda)^2}+2\right)\mathbb{E}_{\boldsymbol{w}}\left[\|\tilde{\boldsymbol{q}}-\hat{\boldsymbol{p}}\|_1\right], \tag{29}$$

where $S^2(\vec{u}) = \frac{1}{|\vec{u}-1|}\sum_{a\in\vec{u}}(a-\bar{a})^2$ and $\bar{\vec{u}} = \frac{1}{|\vec{u}|}\sum_{a\in\vec{u}}a$.

• BOUNDING TERM II

We begin with the two inequalities

$$\mathsf{y}^\mathsf{T}Q^{-1}\mathrm{diag}(Q(\hat{\boldsymbol{p}}-\tilde{\boldsymbol{q}}))Q^{-\mathsf{T}}\mathsf{y} \leq \|Q^{-\mathsf{T}}\mathsf{y}\|^2\|Q(\hat{\boldsymbol{p}}-\tilde{\boldsymbol{q}})\|_\infty$$
$$\leq \|Q^{-\mathsf{T}}\mathsf{y}\|^2|Q|_\infty\|\hat{\boldsymbol{p}}-\tilde{\boldsymbol{q}}\|_1, \tag{30}$$

where $|Q|_\infty = \max_{i,j}|Q_{ij}|$. For every $\ell \in \mathcal{Y}$ $\tilde{q}_\ell \geq \gamma$ (Esfandiari et al. (2022), Theorem 6), and $\sum_{\ell\in\mathcal{Y}}\tilde{q}_l = 1$, which implies that $\forall \ell \in \mathcal{Y}, \tilde{q}_l \leq 1 - (K-1)\gamma$. Therefore, by definition of $Q$, we have

$$|Q|_\infty \leq 1 - \lambda + \lambda(1-(K-1)\gamma) = 1 - \lambda(K-1)\gamma.$$

The following lemma bounds the maximum singular value of matrix $Q$.

**Lemma A.11.** *The maximum singular value of label randomization matrix $Q^{-1}$ is at most $\frac{\lambda\sqrt{K}+1}{1-\lambda}$.*

Using Lemma A.11, we get

$$\mathsf{y}^\mathsf{T}Q^{-1}\mathrm{diag}(Q(\hat{\boldsymbol{p}}-\tilde{\boldsymbol{q}}))Q^{-\mathsf{T}}\mathsf{y} \leq \frac{(\lambda\sqrt{K}+1)^2}{(1-\lambda)^2}\|\mathsf{y}\|^2(1-\lambda(K-1)\gamma)\|\hat{\boldsymbol{p}}-\tilde{\boldsymbol{q}}\|_1. \tag{31}$$

● BOUNDING $\mathrm{Var}(u_{0,c}|c)$

Combining (29) and (31) with the expression of $\mathrm{Var}(u_{0,c}|c, z)$, we get

$$\mathrm{Var}_{DP}(u_{0,c}|z, \mathcal{P})$$

$$= n_{0,c}\left(\underbrace{y^\mathsf{T}\mathbb{E}_{w}\left[Q^{-1}\mathrm{diag}(Q\tilde{q})Q^{-\mathsf{T}}|z,\mathcal{P}\right]y}_{\text{term I}} + \underbrace{y^\mathsf{T}\mathbb{E}_{w}\left[Q^{-1}\mathrm{diag}(Q(\hat{p}-\tilde{q}))Q^{-\mathsf{T}}|z,\mathcal{P}\right]y}_{\text{term II}}\right) - \sum_{i\in\mathcal{O}_{0,c}} y_i^2(0)$$

$$\leq (n_{0,c}-1)\frac{S^2(\vec{y}_{0,c})}{(1-\lambda)^2} + n_{0,c}\left((\overline{\vec{y}_{0,c}})^2 + B^2\left(\frac{3}{(1-\lambda)^2} + 2\right)\mathbb{E}_{w}\left[\|\tilde{q}-\hat{p}\|_1\right] + \right.$$

$$\frac{(\lambda\sqrt{K}+1)^2}{(1-\lambda)^2}\|y\|^2(1-\lambda(K-1)\gamma)\mathbb{E}_{w}\|\hat{p}-\tilde{q}\|_1\Big) - \sum_{i\in\mathcal{O}_{0,c}} y_i^2(0)$$

$$= \frac{n_{0,c}-1}{(1-\lambda)^2}S^2(\vec{y}_{0,c}) + n_{0,c}(\overline{\vec{y}_{0,c}})^2 - \sum_{i\in\mathcal{O}_{0,c}} y_i^2(0) + n_{0,c}A'_{0,c}\mathbb{E}_{w}[\|\tilde{q}-\hat{p}\|_1]$$

$$= (n_{0,c}-1)\left(\frac{1}{(1-\lambda)^2} - 1\right)S^2(\vec{y}_{0,c}) + n_{0,c}A'_{0,c}\mathbb{E}_{w}[\|\tilde{q}-\hat{p}\|_1], \tag{32}$$

with

$$A'_{0,c} := B^2\left(\frac{3}{(1-\lambda)^2} + 2\right) + \frac{(\lambda\sqrt{K}+1)^2}{(1-\lambda)^2}\|y\|^2(1-\lambda(K-1)\gamma).$$

The final step is bounding the term $\mathbb{E}_{w}[\|\tilde{q}-\hat{p}\|_1]$.

**Lemma A.12.** *Recall the notation $\tilde{q} = [\tilde{q}_0(y|c)]_{y\in\mathcal{Y}}$ and $\hat{p} = [\hat{p}_0(y|c)]$. Then,*

$$\mathbb{E}_{w}[\|\tilde{q}-\hat{p}\|_1] \leq 2K\left[\gamma + \frac{\sigma}{n_{0,c}}\left(e^{-\gamma n_{0,c}/\sigma} - e^{-n_{0,c}/\sigma}\right)\right].$$

By using Lemma A.12 in (32), we obtain

$$\mathrm{Var}_{DP}(u_{0,c}|z, \mathcal{P}) \leq (n_{0,c}-1)\left(\frac{1}{(1-\lambda)^2} - 1\right)S^2(\vec{y}_c(0)) + n_{0,c}A(n_{0,c}), \tag{33}$$

with

$$A(x) = A'_{0,c}2K\left[\gamma + \frac{\sigma}{x}\left(e^{-\gamma x/\sigma} - e^{-x/\sigma}\right)\right]$$

$$= 2K\left(B^2\left(\frac{3}{(1-\lambda)^2} + 2\right) + \frac{(\lambda\sqrt{K}+1)^2}{(1-\lambda)^2}\|y\|^2(1-\lambda(K-1)\gamma)\right)\left[\gamma + \frac{\sigma}{x}\left(e^{-\gamma x/\sigma} - e^{-x/\sigma}\right)\right]$$

A similar bound can be derived for $\mathrm{Var}(u_{1,c}|c)$, which in conjunction with (22) gives the desired result.

**Proof of Lemma A.10**

We recall the statement of Lemma A.10 below for convenience.

$$\mathbb{E}_{\tilde{q},w}[y_i^2(0)]|z, \mathcal{P}] \leq B^2\mathbb{E}_{w}\|\tilde{q}-\hat{p}\|_1 + \frac{1}{n_{0,c}}\sum_{i\in\mathcal{O}_{0,c}} y_i(0)^2. \tag{34}$$

$$\mathbb{E}_{\tilde{q},w}[y_i(0)|z, \mathcal{P}]^2 \geq \left(\frac{1}{n_{0,c}}\sum_{i\in\mathcal{O}_{0,c}} y_i(0)\right)^2 - 2B^2\mathbb{E}_{w}\|\tilde{q}-\hat{p}\|_1. \tag{35}$$

*Proof.* Since the outcomes are bounded by $B$, we have

$$\left|\mathbb{E}_{\tilde{q},w}[y_i^2(0)|z,\mathcal{P}] - \mathbb{E}_{\hat{p}}[y_i^2(0)]\right| = \left|\mathbb{E}_w\left[\sum_{y\in\mathcal{Y}} y^2(\tilde{q}_y - \hat{p}_y)\Big|z,\mathcal{P}\right]\right|$$

$$\leq \mathbb{E}_w\left[\left|\sum_{y\in\mathcal{Y}} y^2(\tilde{q}_y - \hat{p}_y)\right|\ \Big|z,\mathcal{P}\right] \leq B^2\mathbb{E}_w\|\tilde{q} - \hat{p}\|_1.$$

Therefore,

$$\mathbb{E}_{\tilde{q},w}[y_i^2(0)|z,\mathcal{P}] \leq \mathbb{E}_{\hat{p}}[y_i^2(0)|z,\mathcal{P}] + B^2\mathbb{E}_w\|\tilde{q} - \hat{p}\|_1.$$

We next note that

$$\mathbb{E}_{\hat{p}}[y_i^2(0)|z,\mathcal{P}] = \mathbb{E}\left[\sum_{y\in\mathcal{Y}}\sum_{i\in\mathcal{O}_{0,c}} \frac{\mathbb{I}(y_i = y)}{n_{0,c}}y^2\Big|z,\mathcal{P}\right] = \frac{1}{n_{0,c}}\sum_{i\in\mathcal{O}_{0,c}} y_i(0)^2.$$

This completes the proof of (34). Likewise we have

$$\left|\mathbb{E}_{\tilde{q},w}[y_i(0)]^2 - \mathbb{E}_{\hat{p}}[y_i(0)]^2\right| = \left|\mathbb{E}_{\tilde{q},w}[y_i(0)] - \mathbb{E}_{\hat{p}}[y_i(0)]\right| \cdot \left|\mathbb{E}_{\tilde{q},w}[y_i(0)] + \mathbb{E}_{\hat{p}}[y_i(0)]\right|$$

$$\leq 2B\left|\mathbb{E}_{\tilde{q},w}[y_i(0)] - \mathbb{E}_{\hat{p}}[y_i(0)]\right|$$

$$\leq 2B^2\mathbb{E}_w\|\tilde{q} - \hat{p}\|_1.$$

Therefore, we obtain:

$$\mathbb{E}_{\tilde{q},w}[y_i^2(0)|z,\mathcal{P}] \geq \mathbb{E}_{\hat{p}}[y_i(0)]^2 - 2B^2\mathbb{E}_w\|\tilde{q} - \hat{p}\|_1.$$

We next note that

$$\mathbb{E}_{\hat{p}}[y_i(0)]^2 = \left(\sum_{y\in\mathcal{Y}}\sum_{i\in\mathcal{O}_{0,c}} \frac{\mathbb{I}(y_i = y)}{n_{0,c}}y\right)^2 = \left(\frac{1}{n_{0,c}}\sum_{i\in\mathcal{O}_{0,c}} y_i(0)\right)^2.$$

This completes the proof of (35). □

## Proof of Lemma A.11

**Lemma A.11.** *The maximum singular value of label randomization matrix $Q^{-1}$ is at most $\frac{\lambda\sqrt{K}+1}{1-\lambda}$.*

*Proof.* For any unit norm vector $u$ we have

$$u^\mathsf{T}Q^{-1} = \frac{1}{1-\lambda}u^\mathsf{T} - \frac{\lambda}{1-\lambda}u^\mathsf{T}\tilde{q}\mathbf{1}^\mathsf{T}.$$

Therefore, by triangle inequality

$$\|u^\mathsf{T}Q^{-1}\| \leq \frac{1}{1-\lambda} + \frac{\lambda}{1-\lambda}\|\tilde{q}\| \cdot \|\mathbf{1}\| \leq \frac{1+\lambda\sqrt{K}}{1-\lambda},$$

where in the last step we used $\|u\| = 1$ and $\|\tilde{q}\| \leq \|\tilde{q}\|_1 = 1$. □

**Proof of Lemma A.12**

**Lemma A.12.** *Recall the notation $\tilde{\boldsymbol{q}} = [\tilde{q}_0(y|c)]_{y \in \mathcal{Y}}$ and $\hat{\boldsymbol{p}} = [\hat{p}_{0,D}(y|c)]$, where we dropped the subscript c to lighten the notation. Then,*

$$\mathbb{E}_{\boldsymbol{w}}[\|\tilde{\boldsymbol{q}} - \hat{\boldsymbol{p}}\|_1] \leq 2K\left[\gamma + \frac{\sigma}{n_{0,c}}\left(e^{-\gamma n_{0,c}/\sigma} - e^{-n_{0,c}/\sigma}\right)\right].$$

We follow the proof of (Esfandiari et al. (2022), Lemma 5). By a tighter derivation which carries over in a straightforward way, we obtain the following bound analogous to Equation (6) therein:

$$\mathbb{E}_{\boldsymbol{w}}[\|\tilde{\boldsymbol{q}} - \hat{\boldsymbol{p}}\|_1] \leq 2\sum_{y \in \mathcal{Y}} \mathbb{E}_{\boldsymbol{w}}[\max(\gamma, \min(1, |w_{y,c}|))] = 2K\mathbb{E}[\max(\gamma, \min(1, V))],$$

where $V = |w_{y,c}| \sim \text{Exp}(n_{0,c}/\sigma)$, since $w_{y,c} \sim \text{Laplace}(\sigma/n_{0,c})$. For a random variable $V \sim \text{Exp}(\alpha)$, we have

$$\begin{aligned}
\mathbb{E}[\max(\gamma, \min(1, V))] &= \int_0^\gamma \gamma \alpha e^{-\alpha v} \mathrm{d}v + \int_\gamma^1 v \alpha e^{-\alpha v} \mathrm{d}v + \int_1^\infty \alpha e^{-\alpha v} \mathrm{d}v \\
&= -\gamma e^{-\alpha v}\Big|_0^\gamma - (v + \frac{1}{\alpha})e^{-\alpha v}\Big|_\gamma^1 - e^{-\alpha v}|_1^\infty \\
&= \gamma + \frac{1}{\alpha}(e^{-\alpha \gamma} - e^{-\alpha}),
\end{aligned}$$

which after substituting for $u = n_{0,c}/\sigma$ gives the claim.

You can have as much text here as you want. The main body must be at most 8 pages long. For the final version, one more page can be added. If you want, you can use an appendix like this one.

The `\onecolumn` command above can be kept in place if you prefer a one-column appendix, or can be removed if you prefer a two-column appendix. Apart from this possible change, the style (font size, spacing, margins, page numbering, etc.) should be kept the same as the main body.

