# OpenReview forum: "Improving the Variance of Differentially Private Randomized Experiments through Clustering"
_ICML.cc/2025/Conference — ICML 2025 poster_

### Official Review · Reviewer_WorV · 2025-03-13

**Overall Recommendation:** 3

**Summary:**

This paper proposes a differentially private algorithm for causal effect estimation, which leverages cluster structure in the data in order to reduce the variance (i.e., improve utility) while maintaining the same privacy guarantee.

## update after rebuttal

I’m bumping up my score, after reading the rebuttal and the other reviews. I think the novelty and theoretical results are strong selling points of the paper, though I do still feel that the label DP setting limits its scope. I think the paper would benefit from a more careful exposition of label DP (including its limitations and what it does and does not protect) and also in what scenarios is it reasonable to assume that attributes are non-sensitive.

**Claims And Evidence:**

The claims made in the submission are supported by evidence.

**Essential References Not Discussed:**

All relevant related works are discussed in the paper, as far as I can tell.

**Experimental Designs Or Analyses:**

I didn't check that carefully, but the experimental designs looked sound to me.

**Methods And Evaluation Criteria:**

The methods make sense for the problem at hand.

**Other Comments Or Suggestions:**

I might have liked to see more justification on why label DP makes more sense as a privacy setting compared to standard DP.

**Other Strengths And Weaknesses:**

Strengths --
- Novel premise: I appreciated the idea of leveraging cluster structure in order to improve utility.
- I thought the theoretical results were quite nice — presented cleanly, technically sound (to the best of my knowledge) and interpretable. For example, Theorem 3.4 is a nice result that provides insight into what type of clusters can best reduce the variance.
- The paper is written nicely and communicates its ideas effectively.

Weaknesses --
- The proposed algorithm is very heavily inspired by one particular application (descibed in detail on the first page). It’s not totally clear to me how well this approach would generalize to other applications. In particular, any other use case for CLUSTER-DP would require access to non-sensitive user attributes which might not always be readily available. (The authors did address this point in Remark 2.1, but I’m not sure I’m wholly satisfied.)
- The experimental results are mostly based on a numerical simulation; assuming I haven’t missed anything, the only non-simulated dataset is the Youtube dataset. To demonstrate the practicality of the algorithm, it would have been nice to see more experiments conducted on a broader array of realistic datasets.

**Questions For Authors:**

- Going back to Remark 2.1, let’s say that the cluster structure is sensitive and privatizing the cluster requires $\epsilon_1$ privacy budget. For the end-to-end process to have the same privacy guarantee $\epsilon  = \epsilon_1 + \epsilon_2$ as the algorithm with non-sensitive cluster structure, we’d need reduce the $\epsilon_2$ budget for privatizing the outcomes and thus re-introduce more variance. I am wondering if it would be possible to address this point? For example, are there allocations of $\epsilon_1$ and $\epsilon_2$ for which the end-to-end process would still have less variance than the non-sensitive cluster situation where the privacy budget can be fully devoted to privatizing the outcomes (i.e., $\epsilon = \epsilon_2$)?

- Another question that I have about Remark 2.1 is that wouldn't the end-to-end process mix DP (for privatizing the clusters) and label DP (for privatizing the outcomes)? Would the resulting privacy guarantee thus be possible to easily interpret?

- The paper’s approach is to set a privacy level, then for certain well-behaved cluster structures be able to achieve better utility. Rather than showing improved utility via bounding the variance of the estimator, I would be interested to know if the authors considered using something like Propose-Test-Release to reduce the noise scale (and maybe more directly improve the utility of the algorithm) for nicely-behaved clusters.

**Relation To Broader Scientific Literature:**

The contributions of the paper are related to differentially private causal inference.

**Theoretical Claims:**

I didn't check the correctness of any of the proofs.

---

> ### Author Rebuttal · Authors · 2025-03-31
>
> We would like to thank the reviewer for their careful review of our paper. We hope to have addressed their questions, and would be happy to clarify these points in a camera-ready version.
>
> _Response to weaknesses:_
>
> Although we focus the presentation on a motivating application in the advertising space—which is already a very broad and important application, deriving most of the revenue of big tech companies—this framework can definitely be applied to other domains. We discuss a few here:
> - Healthcare: To analyze the causal effect of medical interventions, treatment strategies, and healthcare policies while preserving patient privacy. Here, the clustering can be done based on non-sensitive information, e.g. age, race, demographic information.
> - Finance: To study the causal impact of economic policies and investment strategies on financial outcomes, risk management, and market stability. Here, the clustering can be done based on the company industry classification information (eg. NAICS), size of the company, other public data (revenue, profit, stock price,...)
> - Social Sciences: To analyze the causal impact of social policies and societal factors on users behavior. Here, the clustering can be done demographic information as well as other publicly available data on users' social platforms.
>
> _Response to Questions:_
>
> - The variance of the full DP case will be larger than the partial DP (non-sensitive cluster and private response).  Since $\epsilon_2 <\epsilon$, in the full DP we need to add more noise in randomizing the responses. In addition, to preserve the privacy of clusters, we need to have some randomization there as well. This affects the variance because it  increases the cluster homogeneity quantities $\phi_0,\phi_1$, since these will now be defined w.r.t noisy clusters and will naturally  be larger than when they are defined w.r.t actual clusters.
> - No, it wouldn’t mix. In fact two important properties of differential privacy (which are widely used in practice and design) are composition and post-processing. The first states that combining two DP mechanisms will remain DP (but their privacy loss will be added); see sec 3.5 in [1]. The latter states that DP is immune to post-processing; see prop 2.1 in [1].
> - The propose-test-release (PTR) mechanism aims to reduce the noise addition for privacy by working with local sensitivity instead of the global sensitivity. Such an approach can be seamlessly integrated into our mechanism, specifically in Algorithm 1 (line 228-229), when adding the Laplace noise: we can scale its noise using PTR and local sensitivity of empirical distribution of the cluster. However, we believe this would yield only minor benefits since the global sensitivity of empirical distribution is already 1/cluster-size; the PTR works most effective when the range of outputs is large.
>
> [1]: “The Algorithmic Foundations of Differential Privacy”, Cynthia Dwork, Aaron Roth

---

### Official Review · Reviewer_iSPq · 2025-03-13

**Overall Recommendation:** 4

**Summary:**

Authors give an algorithm they call Cluster-DP, which is a pure/approximate DP mechanism (label DP) for causal effect estimation. Its main insight is that you can reduce the variance of the estimates by leveraging known clustering structure in the data. At a high level, they add Laplace noise to the empirical response distributions within each cluster, do some clever truncation and renormalization, and get a data-dependent response distribution. They construct a response randomization matrix (again, novelty) and then use its inverse to debias the privatized outcomes, estimating average treatment effects. The main contributions are theoretical guarantees for the algorithm (label differential privacy guarantees and a detailed bound on the estimator’s variance gap relative to a non-private baseline). Their analysis carefully shows how improved cluster homogeneity leads to reduced variance, thereby achieving a better privacy-variance trade-off. They have some empirical evaluations on both synthetic and real data which show the usefulness of their approach in low-epsilon regimes, which is unsurprising following the bounds they give, but good validation.

**Claims And Evidence:**

Overall, I found that the authors made some novel and well-motivated claims, and managed to provide ample evidence (mainly theoretical) to support their Cluster-DP algorithm. The detailed derivations in the appendix were appreciated. They also have some empirical performance claims, which are less substantial but adequately validate their theory in my opinion.

As my review is mainly positive, I leave most of it as questions for the authors, to help clarify their work and its presentation.

I'd like to highlight that I found the debiasing step via the inversion of the response randomization matrix $Q_{c,a}$ to be really interesting and non-trivial. However, its stability (especially when $\lambda$ is close to 1) is not formally explored (as far as I could tell) and might be a source of unnecessary variance if the matrix is near-singular (is that right?). Could the authors discuss this a bit further? Extrapolating from the $\lambda$ experiments seem to suggest this could be a problem, but maybe I'm over-interpreting these results.

**Essential References Not Discussed:**

Not aware of any.

**Experimental Designs Or Analyses:**

I appreciated the conditional bias plots and variance gap plots, and the qq-plots. The youtube data seems reasonable for this problem. I'd be curious for the authors to comment on the empirical efficiency of their algorithm?

I was unable to carefully check their experiments as the code wasn't provided.

**Methods And Evaluation Criteria:**

The authors do not provide an extensive empirical evaluation, only a standard evaluation with a synthetic dataset and on (real) youtube data. This is adequate, as the main results of the paper are theoretical. However, more in depth empirical evaluation of Cluster DP would be great in future work.

**Other Comments Or Suggestions:**

Nitpicks (mostly notational):
Maybe I missed something, but the vector $y$ is used to denote the entire response space and in expressions like $\mathbf{y}^\top$. Can you distinguish between the vector of outcomes and the set $\mathcal{Y}$ of possible outcomes in your notation?

Some proofs switch between notations such as $\mathbf{y}_{0,c}^2$ and "\overline{y^2_{0,c}}"  $= 1 / n_{0,c} \sum y_i^2 (0)$ or something, which made it hard for me to follow whether it was a sum or an average sometimes. Some of the $\bar{y}$ and $\overline{y_{0,c}}$ notational norms were more confusing than helpful.

You left some of the text from the template at the very bottom of the appendix, please delete in camera ready.

**Other Strengths And Weaknesses:**

Strengths:
I congratulate the authors on this work, which I think is original and clearly presented. That said, it's not clear to me that it is particularly "significant" in that I'm not as aware of the desire for label-DP privacy guarantees in clustering for downstream causal effect estimation (a little niche).

So, in that sense, I would not consider this work to be award level, but I certainly believe it merits acceptance.

Weaknesses:
The authors do a good job motivating the work in the intro for a specific application.

However, the intuition for their approach could be better communicated. I found myself drawing some simple pictures to try to get a sense of why the Cluster-DP algorithm is a good idea for variance reduction; the authors should provide better intuition (through plots or a carefully constructed distributional example) to help the reader grasp the approach.

**Questions For Authors:**

Q1. Theorem 3.1 presents a privacy bound that splits into a term derived from the Laplace mechanism and a term from the re-sampling process. Could you please discuss whether these bounds are tight in practice (I'm not sure your experiments really give us a sense, but maybe I'm missing something)? Are there settings or data regimes where you'd expect the analysis to be overly conservative?

Q2. Your estimator relies on inverting the response randomization matrix $Q_{c,a}$. Under what conditions might this matrix become ill-conditioned, and how does that affect the variance of $\hat{\tau}_Q$? Is there a safeguard in your method for cases where the inversion is unstable? Maybe I'm misunderstanding something, I'd appreciate if the authors could help me here.

Q3. My understanding of both the theory and the empirical results is that the variance reduction benefits rely on cluster homogeneity (authors parameterize this as $\phi_a$). Can you elaborate on how robust your method is to mis-specified or sub-optimal clustering? E.g., if clusters are more heterogeneous than assumed? This is sort of touched on by considering "Cluster-free DP," but I'm wondering if there's a smooth interpolation between that and the "Cluster-DP" version.

Q4. This is a future work question: you gave a mechanism that, as far as I can tell, seems to work only for discrete outcome spaces (even though you expand it beyond binary). Is there hope for continuous outcomes? Can you discuss the foreseeable issues more carefully? You mention binning - the bin sizes could be parameterized, and done in such a way that adapts to the cluster structure. This seems natural, and may be worth discussing in future work.

**Relation To Broader Scientific Literature:**

The manuscript appears well-situated within the literature. In particular, the authors claim a "tighter analysis" [Esfandiari et al. 2022] than previous work and substantiate the claim.

**Theoretical Claims:**

I congratulate the authors on their careful formal statements and detailed proofs. I checked the proofs of Theorems A.5 (on the baseline uniform approach) and 3.1 (the Cluster DP mechanism). I did not find issues with either. Though I did not check it carefully, I skimmed the proof of Theorem 3.4 (for the main variance reduction claim), which is well broken up into propositions and lemmas.

As a general statement on the theory in this paper, though many of the tools used are standard, they are carefully and effectively combined in non-trivial ways, and the theory is broken up nicely to construct the desired bounds.

---

> ### Author Rebuttal · Authors · 2025-03-31
>
> We thank the reviewer for their careful and positive review of our paper. We would be happy to clean up the notational remarks made by the reviewer, and clarify the points below in a camera-ready version. We now address their questions:
>
> _Q1._
>
> In our proof of Theorem 3.1, we use the composition theorem of differential privacy. In general, if mechanisms Mi are $(\epsilon_i,\delta_i)$- DP, then the composition satisfies $(\sum_i \epsilon_i, \sum_i \delta_i)$ DP. We use these results with $M_1$ the Laplace mechanism and $M_2$ the re-sampling process. If we do not allow any slack in the failure probability $\delta$, then this cannot be tightened, i.e., one can find examples of mechanisms which violate $(\epsilon, \sum_i \delta_i)$ DP if $\epsilon< \sum_i \epsilon_i$; see [1]. But as shown in [1] if we allow for a larger value of $\delta$, one can improve the privacy in terms of $\epsilon$. This tightness is for general mechanisms. For the specific composition of Laplace mechanism and resampling, there may be some tighter bounds, but an analysis based on the composition property (as is our argument) cannot be tightened. Given that, we do not anticipate the bound to be excessively conservative, if conservative at all.
>
> [1] “The Composition Theorem for Differential Privacy”, Peter Kairouz et al, 203.
>
>
> _Q2._
>
> This is a good question. The matrix $Q_{c,a}$ is a scalar of the identity matrix plus a low rank matrix. As shown in the supplementary material (lines 1698-1704), its inverse is also a rank one perturbation of the (scaled) identity matrix. In Lemma A.11 we bound the maximum eigenvalue of $Q^{-1}$, which is the inverse of the minimum eigenvalue of $Q$. Using this lemma, the minimum eigenvalue of $Q^{-1}$ is at least $(1-\lambda)/(\lambda\sqrt{K}+1)$. So the matrix is well-conditioned if $\lambda$ (probability of resampling in the DP mechanism) is strictly less than 1, which is the case in our algorithm (by choosing $\sigma< \epsilon$ in privacy bound Thm 3.1).
> This is also reflected in the variance bound, Thm 3.4, where $(1-\lambda)^2$ appears in the denominator.
>
> _Q3._
>
> Thank you for raising this question. Our privacy guarantees are entirely robust to the chosen clusters! The statements of Theorem 3.1 and its corollaries do not depend on the cluster structure and properties; only the variance gap established in Theorem 3.4 depends on the cardinalities and homogeneity of clusters. In practice, clusters being more heterogeneous than assumed would show up as increased variance, which is measurable, but this would not endanger the integrity of the privacy claims. We do empirically evaluate the role of clustering quality in Experiment 2 and shown in Figure 1.c by varying beta which directly controls the cluster homogeneity (larger beta corresponding to more homogeneous clusters). As expected the performance of our results (in terms of variance gains over the Cluster-Free algorithm) improves at larger betas, and smaller lambdas (the resampling probability).
>
> _Q4._
>
> Absolutely. This is a great point, and definitely an exciting research direction for future work. As you noted, we propose binning as a solution, and implement it in the Numerical Experiments of Section 4, for both the fully synthetic and semi-synthetic data. We find the results satisfactory despite no particular tuning for the bin number and sizes. Using an adaptive grid is a natural approach to examine, which could yield further gains in the resulting privacy-variance tradeoff. Some immediate issues that practitioners should watch out for are 1) our guarantees depend on the number of bins K, hence one cannot choose an infinite number of bins, and 2) a data-adaptive choice of bin sizes requires access to the dataset which could leak information about the users’ data. This requires calibrating the noise level to ensure the privacy guarantee is maintained.

---

> > ### Comment · Reviewer_iSPq · 2025-04-01
> >
> > I appreciate your careful answers to my questions, thank you. In particular, thank you for the care you took in answering (Q2) - this might be worth sketching out for a reader somewhere in your paper body, but I think I understand now.
> >
> > I was surprised that the other reviewers were less positive. I will maintain my score, best of luck.

---

> > > ### Author Response · Authors · 2025-04-07
> > >
> > > Thank you for your positive comments on our work! We also responded in details to other reviewers and were hoping that they increase their score, but unfortunately we didn't hear from them after our rebuttal.

---

### Official Review · Reviewer_LKUF · 2025-03-14

**Overall Recommendation:** 3

**Summary:**

This paper introduces Clustered-DP, a differentially private mechanism designed to improve the privacy-variance trade-off in randomized experiments. The proposed method improves the variance-precision trade-off compared to the traditional method, which introduces noise to the sensitive variables.
The paper provides theoretical privacy guarantees under label differential privacy and derives bounds on the variance of the causal effect estimator.

**Claims And Evidence:**

One of the paper's key contributions is that the proposed Cluster-DP approach improves the privacy-variance trade-off. Theorems 3.1 and 3.4 provide the theoretical guarantee of privacy and variance, respectively. In section 4, both simulated and real-world data are used to demonstrate that CLUSTER-DP achieves lower variance compared to baseline methods while preserving privacy guarantees.

**Essential References Not Discussed:**

I'm not aware of essential references that have not been discussed.

**Experimental Designs Or Analyses:**

The paper uses both synthetic and real-world datasets to evaluate performance and demonstrate the proposed method's generalizability. I checked experiments with numerical and real-world data. I think the results and analyses reflect the paper's key contribution.

**Methods And Evaluation Criteria:**

The CLUSTER-DP algorithm is well-designed to balance privacy and variance, leveraging non-sensitive cluster structures. The experiments show the comparisons to the baseline methods such as uniform-prior-dp.  The evaluation criteria show the privacy-variance trade-off, which is the main focus of the paper.

**Other Comments Or Suggestions:**

(Please see the question below.)

**Other Strengths And Weaknesses:**

- The paper is overall well-written, with clear motivation and rigorous derivations to the main results.

- The proposed method is demonstrated with both synthetic and real-world datasets.

- Although the choice of truncation parameter is discussed in the experiment section, it seems unclear how to choose the parameter in general.

**Questions For Authors:**

- Could you give some more detail about the derivation from l.1169 to l.1177?

- Could you provide some intuition about the property $\tilde{q}_a(y|c) \geq \gamma$ ?

**Relation To Broader Scientific Literature:**

The paper is closely related to prior work on differential privacy and causal inference. It expands on existing methods by introducing clustering-based variance reduction.

**Theoretical Claims:**

I checked the proof of Theorem 3.1 and didn't spot any errors or issues in the arguments. I didn't look at the results that are used in the references, and there are some derivations I didn't understand.

---

> ### Author Rebuttal · Authors · 2025-03-31
>
> We thank the reviewer for their careful and overall positive review. We address below their two questions:
>
> _Q1._
>
> These derivations aim to establish the differential privacy guarantee of a mechanism $M_2$ which resamples labels at random with probability $\lambda$ from the true distribution or a perturbed distribution $\tilde q$.
>
> L. 1170 states the probabilities that the “privatized” outcome $\tilde y$ is equal to a given potential outcome y. If y is the true potential outcome, then either it was sampled directly with probability (1-lambda) or it was sampled from $\tilde q$ with probability $\lambda$. Similarly, if $y$ is NOT the true potential outcome, then it must have been sampled from $\tilde q$ with probability $\lambda$.
>
> L. 1171-1173 identifies the necessary and sufficient conditions for proving that the mechanism $M_2$ is (\epsilon, \delta)-DP.
>
> We now detail the steps made on L. 1174-1177  in further detail, starting with substituting the event probabilities identified on L. 1170.
>
> Step 1 (substituting event probabilities): $1 - \lambda + \lambda \tilde q(y) \leq e^{\tilde \epsilon} (\lambda \tilde q(y)) + \delta$
>
> Step 2 (rearranging terms): $1 - \lambda + \lambda \tilde q(y) ( 1- e^{\tilde \epsilon)) \leq \delta$
>
> By the definition of $\delta := \max(0, 1 -\lambda + \lambda \gamma ( 1- e^{\tilde \epsilon}))$, this condition is equivalent to showing two inequalities:
>
> (a) $1 - \lambda + \lambda \tilde q(y) ( 1- e^{\tilde \epsilon}) \leq 0$ and
>
> (b) $1 - \lambda + \lambda \tilde q(y) ( 1- e^{\tilde \epsilon}) \leq 1 -\lambda + \lambda \gamma ( 1- e^{\tilde \epsilon})$
>
> (b) always holds because $\tilde q(y) \geq \gamma$ and $1- e^{\tilde \epsilon} \leq 0$ since $\tilde \epsilon \geq 0$.
>
> Therefore the initial condition on L. 1171 is equivalent to inequality (a), which can be rearranged to the form see on L. 1177: $0 \leq \lambda (\gamma - \tilde q(y)) ( 1 - \tilde \epsilon)$
>
> _Q2._
>
> We enforce the property that $\tilde q_a(y|c)$ as an initial step of Algorithm 1. This is necessary to obtain the differential privacy guarantee in Theorem 3.1. The intuition is as follows: if for a given cluster $c$, treatment assignment $a$, and potential outcome $y$, we have  $\tilde q_a(y|c) = 0$, then observing $y$ as a “privatized” output for that cluster and treatment assignment would mean that $y$ was equal to the original non-privatized outcome of individual $i$. This would be a violation of the differential privacy principle which aims to provide plausible deniability to individuals: seeing the output shouldn't allow you to be certain about any single individual's data.
>
> We thank the reviewer for their patience with what is admittedly a notational-heavy subject. We would be happy to include these clarifications in a camera-ready version.

---

### Official Review · Reviewer_giHc · 2025-03-15

**Overall Recommendation:** 3

**Summary:**

The paper proposes CLUSTER-DP, a differentially private mechanism aimed at improving the variance of causal effect estimation in randomized experiments by utilizing clustering structures within data. Traditional differential privacy (DP) approaches introduce noise to protect privacy, resulting in increased estimator variance. To improve this privacy-variance trade-off, the authors introduce clustering to guide noise addition, defining a new measure of "cluster quality" (cluster homogeneity) that quantifies intra-cluster variability of outcomes. They prove that leveraging high-quality clusters (more homogeneous groups) substantially reduces variance penalties compared to unclustered or uniform-prior baselines.

**Claims And Evidence:**

In the motivating application on online advertising, the authors claim that the non-sensitive cluster information can be shared and utilized to improve the mechanism's privacy-variance trade-off. However, this is only correct when the non-sensitive information has no correlation to the private data. Although the DP guarantee (values of \epsilon, \delta) might not change if correlation exists, the privacy is still comprised as the attack can infer the private information through the released clusters. This privacy risk has been quantified and analyzed by inferential privacy [1] and its follow-ups (e.g., [2-4]). The author should discuss this.

[1] Ghosh, Arpita, and Robert Kleinberg. "Inferential privacy guarantees for differentially private mechanisms." arXiv preprint arXiv:1603.01508 (2016).

[2] Song, Shuang, Yizhen Wang, and Kamalika Chaudhuri. "Pufferfish privacy mechanisms for correlated data." Proceedings of the 2017 ACM International Conference on Management of Data. 2017.

[3] Zhang, Wanrong, Olga Ohrimenko, and Rachel Cummings. "Attribute privacy: Framework and mechanisms." Proceedings of the 2022 ACM Conference on Fairness, Accountability, and Transparency. 2022.

[4] Wang, Shuaiqi, et al. "Inferentially-Private Private Information." arXiv preprint arXiv:2410.17095 (2024).

**Essential References Not Discussed:**

See 'Claims And Evidence'

**Experimental Designs Or Analyses:**

Experiments make sense

**Methods And Evaluation Criteria:**

Yes

**Other Comments Or Suggestions:**

NA

**Other Strengths And Weaknesses:**

Strengths:
- The paper is well-organized and easy to follow
- Problem formulation and theoretical analysis are solid
- Numerical experiments are conduct

Please see 'Claims And Evidence' for main weakness

**Questions For Authors:**

- Does the paper assume there is no correlation between the private and non-sensitive information? If not, how to model this correlation?
- In Thm 3.1, $\epsilon$ is related to $1/\gamma$, where $\gamma\leq 1/K$. Can the authors provide intuitions on why $\epsilon$ increases with $K$?

**Relation To Broader Scientific Literature:**

The paper is related to previous DP literatures

**Theoretical Claims:**

The proofs seem correct.

---

> ### Author Rebuttal · Authors · 2025-03-31
>
> We would like to thank the reviewer for taking the time to read our paper and for their thoughtful comments. We would be happy to include clarifications of the points below in a camera-ready version.
>
> *Correlation between non-sensitive and private data:*
>
> It seems that your comment is about the measure of differential privacy and its limitations. We would like to highlight a few points:
> - There is a discussion about this in chapter 1 (The Promise of Differential Privacy) of [1] with an example about a medical study that teaches smoking may be correlated to cancer. Does this study compromise the smoker’s privacy? While the study reveals more information, under differential privacy this information is not deemed to be ‘leaked’. The rationale is that the impact on the smoker is the _same regardless of whether or not he was in the study_. It is the _conclusions reached_ in the study that affect the smoker, not his presence or absence in the data set. In other words, DP focuses on the privacy loss to an individual by her contribution to a dataset and therefore “by design” does not capture all of the privacy losses from correlations.
> - The notion of “inferential privacy” aims to consider privacy leakage due to correlation among individuals (it is identical to DP when individuals’ data are independent). So it is not addressing the case of correlation between sensitive and non-sensitive “features”, rather correlation between “individuals/samples”.
> - In our setting, the features $x_i$ are only used to form the clusters and then both the mechanism and analysis work with responses $y_i$ and cluster memberships $c_i$. Our analysis is under a finite sample setting, but extending it to its super-population equivalent assumes data $(y_i,c_i)$ are i.i.d, but conditional distribution $p(y|c)$ would be different (think of it as users are assigned to clusters independently according to some distribution and the marginal distribution of responses is a mixture distribution.) Notably, the responses $y_i$ are independent and so we don’t think inferential privacy would handle this, as it pertains to settings with correlation among samples.
> - The concern about the correlation between y (response) and c (cluster) is exactly the situation with label DP (where labels are deemed private while features are non-sensitive) and semi-sensitive features [2,3], which we discussed in the paper. Furthermore, in Remark 2.1, we argue that using the composition property of DP, we can extend our work to settings where both clusters and responses are sensitive. This “full DP” setting will also address the privacy leakage from correlations.
>
> We would be happy to add a discussion around these points in the revision.
>
> _[1]: “The Algorithmic Foundations of Differential Privacy”, Cynthia Dwork, Aaron Roth_
>
> _[2]: “Training Differentially private Ad prediction models with semi-sensitive features”, L. Chua et. al_
>
> _[3] “Anonymous learning via look-alike clustering”, A. Javanmard et. al_
>
> *Response to Questions:*
>
> - Please see our response above (third bullet point).  Considering the super-population regime, the correlation between clusters and responses is captured in the conditional distribution $p(y|c)$. We do not make any specific assumptions on it (which speaks to the generality of our framework) but it shows up in _Cluster homogeneity_ (Def 3.3). As discussed below the definition in population regime, $\phi_a = E[Var(y(a|c))]$.
> - Note that $\gamma$ is the truncation parameter (algorithm 1), so that $\tilde{q}_a(y|c)\ge \gamma$. This is the distribution used for randomization, and so $\frac{\tilde{q}_a(y|c)}{\tilde{q}_a(y’|c)} \le \frac{1}{\gamma}$. This ratio is at the heart of DP analysis as it concerns the change of outcome distribution by replacing one user. As $K$ (# of possible outcomes) increases, it forces a bound on $\gamma$, which makes this ratio to grow, leading to larger $\epsilon$. A more high-level intuition is that when $K$ is small, an observed randomized response could be allocated to a larger fraction of users (fixing all other params) and so reidentification risk will be smaller.

---

### Decision · Program_Chairs · 2025-05-01

**Decision:**

Accept (poster)

**Comment:**

The paper examines differentially private causal estimation under the label-DP setting, where features are observed but labels are sensitive, and features can be used to cluster users. The authors introduce a novel estimation mechanism based on these clusters that can improve the privacy-utility tradeoff and analyze the mechanism theoretically and empirically, showing that better cluster quality translates to higher utility for the same privacy level.

After discussion, the reviewers were unanimous in favor of acceptance. They found the paper's problem setting and approach novel and the theoretical results rigorous and insightful, with supporting empirical evidence. The two main critiques were: (1) the problem setting, while significant, was perhaps narrow, and (2) concerns about the label-DP setting. For (1), no reviewer felt this was a major concern, and I believe the strengths of the paper outweigh this concern. For (2), the main concern came from Reviewer giHc in their comments about inferential privacy, and to a lesser extent from Reviewer WorV, who had questions about the label-DP setting. I found that these concerns were well addressed by the rebuttal, and both reviewers raised their scores. In particular, my own view is that the issue of inferential privacy is orthogonal to the choice of DP vs. label-DP and that label-DP is an existing notion that is correctly utilized in this paper. The authors can improve the paper by providing a longer discussion of label-DP and its assumptions and any potential limitations.